# Opposing effects of selectivity and invariance in peripheral vision

Corey M. Ziemba [1,2 ✉] & Eero P. Simoncelli[2,3]

Sensory processing necessitates discarding some information in service of preserving and reformatting more behaviorally relevant information. Sensory neurons seem to achieve this by responding selectively to particular combinations of features in their inputs, while averaging over or ignoring irrelevant combinations. Here, we expose the perceptual implications of this tradeoff between selectivity and invariance, using stimuli and tasks that explicitly reveal their opposing effects on discrimination performance. We generate texture stimuli with statistics derived from natural photographs, and ask observers to perform two different tasks: Discrimination between images drawn from families with different statistics, and discrimination between image samples with identical statistics. For both tasks, the performance of an ideal observer improves with stimulus size. In contrast, humans become better at family discrimination but worse at sample discrimination. We demonstrate through simulations that these behaviors arise naturally in an observer model that relies on a common set of physiologically plausible local statistical measurements for both tasks.

[1] Center for Perceptual Systems, The University of Texas at Austin, Austin, TX, USA. [2] Center for Neural Science, New York University, New York, NY, USA. [3] Flatiron Institute, Simons Foundation, New York, NY, USA. ✉email: ziemba@cns.nyu.edu

Sensory signals are typically transduced and represented in great detail. The human eye, for example, measures light intensities using ~5 million cones (and 100 million rods), each responding at millisecond timescales. The visual system must evaluate and process this enormous flow of data, either acting on it, storing it, or discarding it. Given the limitations of resources for both computation and storage, growing evidence supports the notion that much of the information is discarded, preserving only the small portion that is relevant (or potentially relevant) for current or future behavior.

Theories of coding efficiency provide a partial explanation of this process by positing that these systems are optimized to maximize the extracted and transmitted information about the environment while conserving resources (e.g., metabolic costs, wiring length, number of cells). Such principles have succeeded in accounting for various observations in early sensory systems. For example, the need for efficiency can predict aspects of visual encoding in the retina, whose ~1 million axonal fibers form the optic nerve through which all visual information is transmitted to the brain[1–6].

Once this information reaches cortex, the representation appears to expand (in terms of number of cells). Nevertheless, there is evidence for continued loss of information. This is particularly evident in the perceptual deficits in the peripheral portions of the visual field. The most well known of these is decreased acuity: With increasing eccentricity (distance from the fovea), there is a gradual reduction in the ability to detect high spatial frequencies[7,8]. But there are additional perceptual deficits[9,10], many of which are characterized as visual crowding[11], in which recognition of features, objects or letters becomes more difficult in the visual periphery when surrounded by uninformative clutter. Like the falloff of acuity, the spacing at which these effects occur grows with eccentricity[12–14].

Recent work suggests that these deficits may be explained through a process of statistical summary. Specifically, a number of authors have posited that the visual system extracts a set of local summary statistics from visual images, and discards the details from which these summaries are computed[11,15–18]. Physiologically, such statistical summarization can be loosely associated with the pooling behaviors seen in neurons at different stages of the hierarchy (e.g., the integration over bipolar afferents seen in retinal ganglion cells[19]; the combination over simple cells of differing phase or spatial location seen in V1 complex cells[20], the combination over direction-selective complex cells with different direction preferences seen in MT pattern cells[21]). Nevertheless, it is not obvious why the visual system should discard this information, and coding efficiency (at least, in its simplest form) does not seem to provide an answer.

Here, we show that these peripheral losses are accompanied by a gain: As details are lost through summarization, tasks that rely on summary information can improve, allowing the observer to see the forest for the trees. Inspired by analogous studies of the effect of duration on auditory perception of synthetic sound textures[22], we expose this tradeoff by generating stimuli that are matched according to a model of peripheral summary statistics, but that differ in their details. The ability of human observers to discriminate these stimuli worsens as the size of the presentation window increases—a paradox given that the larger windows provide more information for performing the task. In contrast, discrimination of stimuli with different statistics improves with increased window size. This result is distinct from known crowding phenomena, in which task-irrelevant surrounding stimuli degrade discriminability. However, as with acuity and crowding, we show that these effects scale with eccentricity. Finally, we demonstrate through simulations of a physiologically plausible observer model based on the same summary statistics, that the apparent paradox of opposing effects arises naturally and directly from the computation of statistics over spatially localized regions.

## Results

As a substrate for both our model of peripheral vision and for generating experimental stimuli, we utilized a set of summary statistics previously developed for the representation of visual texture[23]. These are physiologically inspired, and are computed from the rectified and pooled output of oriented bandpass filters, emulating simple and complex cells in V1 (Fig. 1a). Spatially averaged covariances are computed between the outputs of pairs of filters that differ in preferred spatial frequency, position, or orientation (Fig. 1a). These statistics capture image features that are sufficient to account for the appearance of many visual

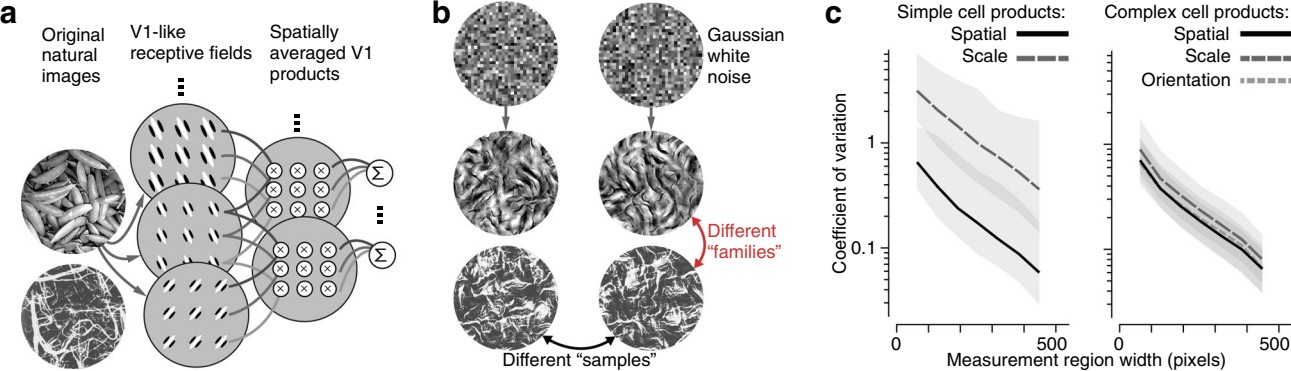

**Fig. 1 Generation of synthetic texture stimuli. a** Original photographs (first column) are decomposed into V1-like responses (rectified oriented linear filter outputs) differing in orientation and spatial frequency (second column). Response statistics are computed by spatially averaging the products of responses at different orientations, spatial frequencies, and local positions (third column)[23]. **b** Images of Gaussian white noise (top) are iteratively adjusted until their statistics match those of the original images. Initializing with different random seeds yields different samples with identical statistics but differing in detail (columns), and different statistics yield samples from different texture families (rows). **c** Statistics converge with increasing measurement region width. Median coefficient of variation (standard deviation across samples divided by mean across samples) of example groups of higher-order statistics, as a function of the width of the region over which the statistics are measured. Products of simple cell responses (linear responses; left) and complex cell responses (magnitude responses; right) were measured from regions cropped from images synthesized to be statistically matched across their full extent (512 × 512 pixels). Shaded regions indicate interquartile range across different individual statistics computed from 15 texture families. Source data are provided as a Source Data file.

textures[23–25]. In addition, they have been used as a basis for explaining visual crowding in the periphery[17,18,26], and are particularly effective in selectively driving responses of V2 and V4 neurons[27–32].

To generate stimuli for testing the effects of summary statistics on peripheral representations, we used the analysis–synthesis procedure developed by Portilla and Simoncelli[23]. First, statistics are measured from a grayscale photograph of a natural texture. A synthetic texture image is generated by iteratively adjusting the pixels of an initially random image until it has the same statistics as the original photograph (Fig. 1a, b). We refer to a set of images generated with identical statistics as a texture "family" (Fig. 1b, different rows). Within a family, initializing the synthesis from different random seeds (Fig. 1b, top) produces images differing in detail, and we refer to all such images as "samples" from that family (Fig. 1b, different columns). By construction, these samples are identical in their model statistics (when measured over the full extent of the image), but differ in the precise location and arrangement of their local features.

Previous work[23–25,33] showed that samples from a given family are similar in appearance to each other, and to the original photograph from which their statistics are drawn, whereas different families are easily differentiated (as can be seen from visual inspection of Fig. 1b). This suggests that the human visual system computes these (or closely related) statistics, and uses them to differentiate texture families[34]. A natural consequence of this hypothesis is that the ability to distinguish different texture families should degrade if the stimuli are spatially cropped, since estimated values of the underlying statistics are less accurate when averaged over a smaller region (Fig. 1c).

An even stronger hypothesis—that the visual system computes these statistics, and then discards the details from which they are computed—leads to an additional and more counterintuitive prediction. The increased variability of texture statistics that arises from spatial cropping implies that our ability to distinguish different samples from the same family should improve for smaller images (and conversely, worsen for larger images). Thus, this hypothesis predicts opposing effects of stimulus size on the discrimination of texture families and texture samples. We designed and performed a set of perceptual experiments to directly test these predictions in the visual periphery.

**Effects of stimulus size on discriminability**. We asked observers to discriminate peripherally presented texture patches of varying size. On each trial, the subject fixated on a point on one edge of the screen while three circular texture patches were briefly presented (for 100 ms) sequentially at the same location in the periphery (Fig. 2d). We asked subjects to indicate with a button press whether the first or last texture was different from the other two. In the family discrimination task, subjects had to differentiate samples from two families with different statistics (Fig. 2a). In the sample discrimination task, subjects had to differentiate two different samples from the same family (and thus, by construction, with identical statistics—Fig. 2b). The size of the texture patches was varied across trials, but all patches were placed so as to maintain a distance of four degrees from the fixation point to the nearest edge (Fig. 2c).

Family discrimination improves with stimulus size (Fig. 2f, red line), as can be confirmed by visual inspection of the example stimuli in Fig. 2a. This effect is consistent across stimuli and observers (Supplementary Fig. 1a,b), and is expected, since the larger stimuli provide more information for the task. Specifically, the statistics on which the discrimination is based can be more accurately estimated from larger stimuli (Fig. 1c).

In contrast, performance in the sample discrimination task was good for small stimuli but became progressively worse as stimulus size increased (Fig. 2f, black line). This is counterintuitive (although not unprecedented[35]), given that the larger patches contain more information for performing the task. At a minimum, one might expect that subjects could maintain performance by basing their judgments on only a small spatial portion of the larger stimuli. Instead, it seems that subjects not only fail to make use of the additional information provided in the larger patches, but that its presence prevents them from accessing the portion they use to discriminate the smaller patches. This effect is thus related to crowding, in which irrelevant contextual information interferes with processing of relevant sensory information[11–17], but reveals a more profound impairment in which even relevant sensory information appears to interfere with processing.

In this sequential version of the task (Fig. 2d), subjects had to remember the earlier stimuli in order to perform the comparison. Limitations of memory might necessitate the use of a summary statistic representation, and this would offer an explanation consistent with analogous observations of opposing effects in auditory texture discrimination[22,36]. We wondered whether allowing subjects to make visual comparisons across space rather than time might diminish or eliminate their reliance on summary statistics. To test this, we repeated the experiment with simultaneous spatially displaced stimuli. The subject fixated the center of the screen while three texture patches were presented equidistant from fixation for 300 ms (Fig. 2e). They indicated with a button press which of the three stimuli was the target (again, the image from a unique family in the family discrimination task, or the unique image in the sample discrimination task). The pattern of performance was quite similar to that for the sequential task: family discrimination performance increased (Fig. 2g, red line) and sample discrimination performance decreased (Fig. 2g, black line) with stimulus size (Supplementary Fig. 1c, d). In fact, performance on the sample discrimination task was even closer to chance levels for large stimuli than it was for the sequential task, potentially due to the spatial uncertainty in making comparisons across portions of the visual field[37]. These results indicate that the pattern of opposing effects is a general feature of family and sample discrimination, and is not due to the process of making comparisons across time, as is inherent to auditory texture discrimination[22,36].

**A local summary statistic observer model predicts opposing effects**. How can we explain this pattern of performance? An ideal observer with full access to stimulus details will improve in both tasks as stimulus size (and thus stimulus information) increases. In contrast, as described previously, an observer with access only to the statistical summaries of each stimulus might be expected to exhibit opposing effects in the tasks because of the gradual convergence of the statistics (Fig. 1c). But visual spatial statistics are presumably computed locally, using neurons with receptive fields of limited spatial extent, which may be smaller than that of the stimulus. We wondered whether, and under what conditions, such a representation would lead to the observed perceptual effects.

To this end, we developed an observer model that computes summary statistics within localized pooling regions that grow in size with eccentricity, analogous to the observed receptive field sizes of neurons in early and mid-level stages of the visual hierarchy[18,38,39]. The front end of the model is identical to the analysis model used to generate full-field visual metamers in Freeman and Simoncelli (2011)[18] (Fig. 3a). As in the model we used for stimulus generation, images are first filtered with a set of V1-like, oriented bandpass filters. Statistics, including covariances between responses differing in spatial position, orientation, and

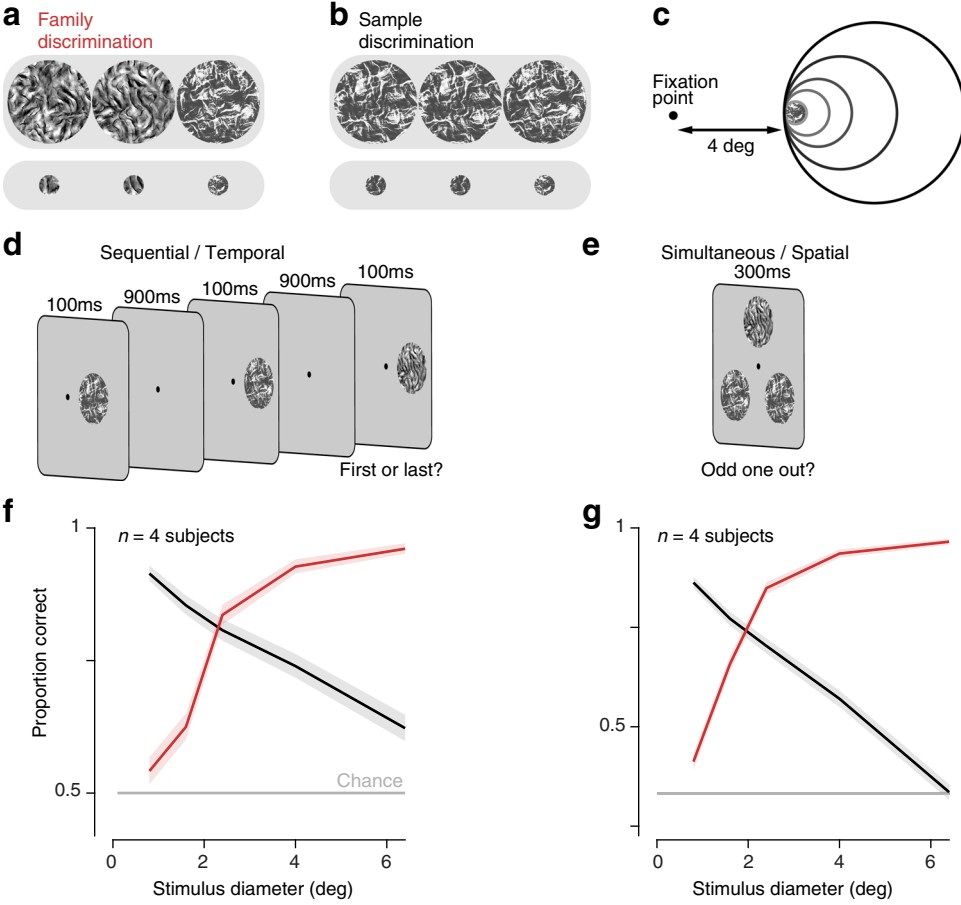

**Fig. 2 Opposing effects of stimulus size on discrimination. a** For family discrimination, each trial consisted of three images, two statistically matched. Subjects were asked to indicate the stimulus from a different category. We presented both large stimuli (top), and smaller stimuli cropped from them (bottom). **b** For sample discrimination, each trial consisted of three statistically matched images, two physically identical. Subjects were asked to indicate the physically different stimulus. **c** Stimuli cropped using circular apertures of five sizes were presented so that the closest edge was 4 degrees from fixation. **d** The sequential/temporal task followed an AXB design (is the middle stimulus different from the first or last?). **e** The simultaneous/spatial task followed an oddity design (which stimulus differs from the other two?). **f** Average proportion correct for four subjects performing the sequential family (red) and sample (black) discrimination tasks as a function of stimulus diameter. Shaded regions represent mean ± SEM across all trials. **g** Average proportion correct for four subjects performing the simultaneous (spatially displaced) task. Shaded regions represent mean ± SEM across all trials. Source data are provided as a Source Data file.

frequency tuning[23], are computed as weighted averages over localized (but smoothly overlapping) pooling regions. We adapted this model output by including normalization and noise to render an observer model for our task. Specifically, each local statistic is divided by the summed activity across all other statistics and pooling regions (Fig. 3a), mimicking physiological gain control effects[40,41]. This induces a form of surround suppression, reducing responses to larger stimuli, and has been recently hypothesized to play a critical role in limiting peripheral discrimination[42]. To quantify discrimination performance, we added independent Gaussian noise to the set of normalized summary statistics, and simulated observer decisions by selecting the image pair with the larger Euclidean distance between these noisy responses.

The architecture and most of the parameters of our model (e.g., oriented filters, choice of statistics) are fixed for all simulations. We examine the effects of varying two free parameters: the rate, $s$, at which the pooling region diameters grow with eccentricity, and the signal-to-noise ratio (SNR) that results from the added noise. Both of these parameters can be related to physiology. Receptive field scaling rates have been previously estimated as approximately $s = 0.21$ for area V1, $s = 0.46$ for V2, and $s = 0.84$ for

V4[18,38,39]. To compute SNR for each $s$ and summary statistic, we take the variance in response across all experimental stimuli and divide by the variance of the added Gaussian noise. The analogous computation applied to single units responding to texture stimuli recorded from anesthetized macaques yields average SNR values of 0.64 in V1 and 0.45 in V2[29]. We therefore assume an SNR of 0.5 to be roughly matched to physiology and simulate model responses with noise levels chosen to yield this average value and a factor of four above and below it.

We generated perceptual predictions by simulating model responses for all experimental stimuli, and compared these to the behavior of the subjects (Fig. 3). In general, we found that family discriminability increased and sample discriminability decreased as stimulus diameter grew, except in the highest SNR cases, for which sample discriminability remained at ceiling (Fig. 3b). At an eccentricity scaling corresponding to V1-sized receptive fields, no value of SNR yielded a good qualitative match with perceptual data (Fig. 3b, top row). Specifically, relative performance between the two tasks did not flip as stimulus size increased. This is presumably because the V1-sized pooling regions of the model are significantly smaller than the receptive fields limiting the performance of human observers. Pooling over small portions of

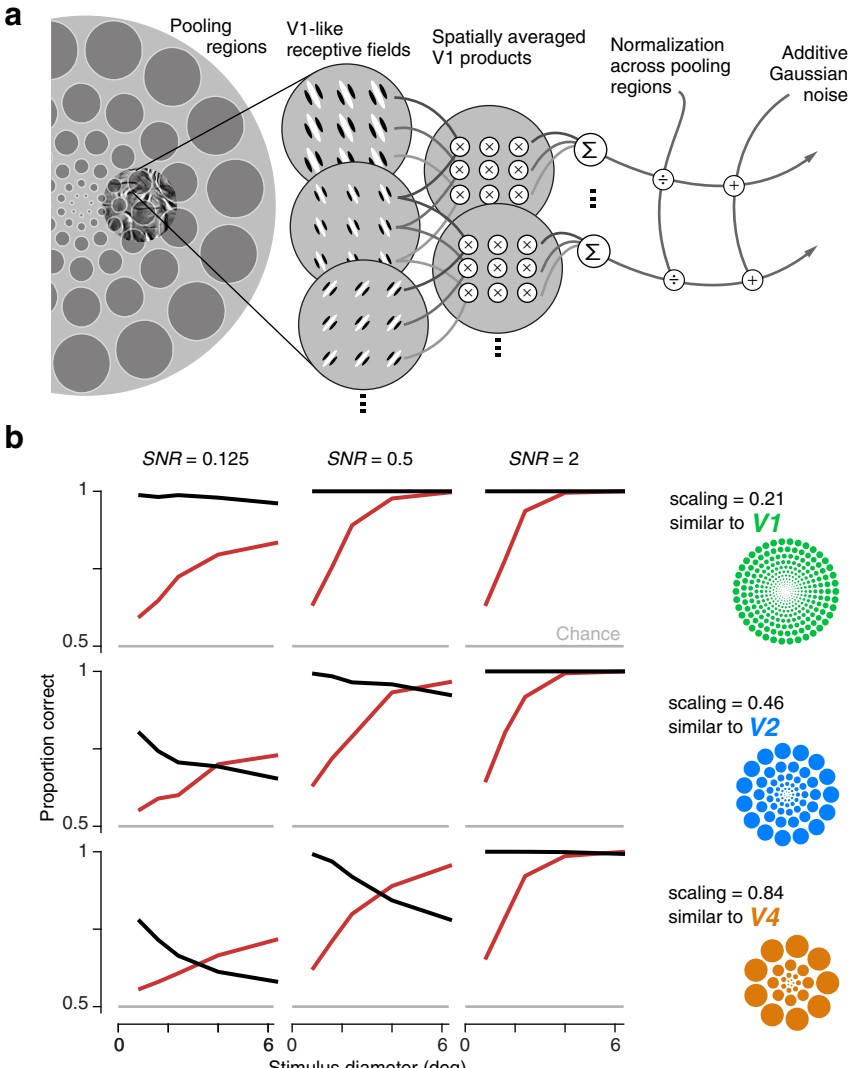

**Fig. 3 A two-stage observer model predicts opposing perceptual effects. a** Starting from V1-like responses (rectified responses of oriented filters), the model computes 668 local pairwise statistics over pooling regions whose sizes grow in proportion to eccentricity (depicted in cartoon form as nonoverlapping discs—actual model responses are computed with smoothly overlapping weighting functions that cover the visual field). Each summary statistic is normalized by the magnitude of responses across all pooling regions, and corrupted by Gaussian noise. The observer response is obtained by computing the Euclidean distance between response vectors for both pairs of images as in the psychophysical AXB design, and selecting the pair with the larger distance. Performance for each condition of noise level, scaling, and stimulus diameter is computed by averaging simulations over 10,000 repeated experiments on 15 samples from four families (as in the perceptual experiments). **b** Performance on family (red) and sample (black) discrimination tasks, as a function of stimulus diameter, for observer models with three different levels of noise and three different scaling values. The average signal-to-noise ratio (SNR) was computed separately for each scaling by dividing the variance in responses across all stimuli and pooling regions by the variance of the Gaussian noise. Scaling values (stimulus diameter divided by eccentricity) are matched to estimates from visual areas V1, V2, and V4. Source data are provided as a Source Data file.

the image yields strongly variable responses across samples, providing strong information for sample discrimination, but poor information for family discrimination. However, increasing the scaling value to that of V2 receptive fields, and assuming physiological SNR levels yielded a pattern of opposing effects resembling the perceptual performance (Fig. 3b, center panel, middle row; compare to Fig. 2f)). Increasing the scaling parameter to that of receptive fields in area V4 further improved the similarity to perceptual performance (Fig. 3b, center panel, bottom row).

We thus conclude that populations of neurons in areas V2 and V4 are likely candidates for the locus of neural selectivity and invariance giving rise to the opposing effects of stimulus size. More generally, these results suggest that the particular summary

statistics used here (correlations across the responses of differently tuned V1 neurons) are represented by populations of V2 and V4 neurons—an idea that is consistent with their computation from V1 afferents and supported by previous psychophysical[18] and physiological[27–32,43] studies.

**Opposing effects scale with eccentricity.** How do these effects change when stimuli are presented at different eccentricities? Because the pooling regions in our observer model scale linearly with distance from the fovea, discrimination performance should be invariant to rescaling of stimuli. That is, a stimulus of a given eccentricity and diameter should have identical discriminability to one at a greater eccentricity if the stimulus diameter is

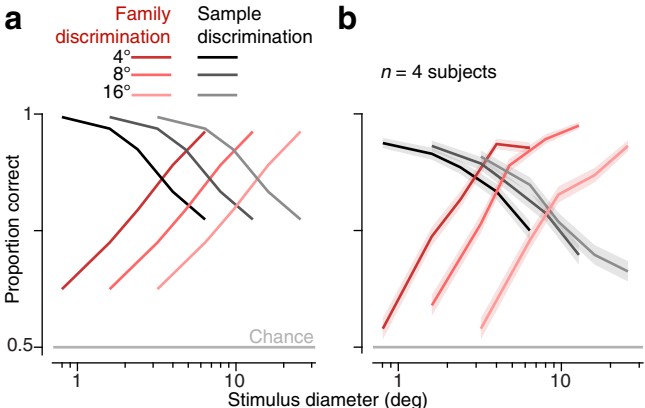

**Fig. 4 Opposing effects scale with eccentricity. a** Observer model performance ($s = 0.84$, SNR $= 0.5$) in family and sample discrimination for different stimulus diameters and eccentricities (note: plotted against log diameter, to facilitate comparisons). **b** Average performance of four subjects performing family and sample discrimination under the same stimulus conditions. Red lines indicate family and black lines indicate sample discrimination. Lighter colored lines indicate larger eccentricities. Shaded regions indicate mean ± SEM across all trials. Source data are provided as a Source Data file.

increased by the same factor[44,45]. We verified this property by computing model performance at eccentricities two and four times those of our original simulations (Fig. 4a, lighter shaded lines). As expected, the curves representing family and sample discrimination performance maintain their shape and relationship to one another, but are shifted rigidly along a logarithmically scaled axis of stimulus diameter (Fig. 4a).

We tested this prediction experimentally by gathering data from human subjects on both tasks at three different eccentricities (4, 8, and 16 degrees). For the two larger eccentricities, we dilated the stimuli by factors of 2 and 4, respectively. This manipulation rescales the spatial frequency content for presentation at different eccentricities, ensuring that stimulus visibility is equated and differing performance across eccentricities is not due to differences in acuity[45]. We found that the pattern of opposing performance across the two tasks shifted, as predicted by our observer model (Fig. 4b)[46], albeit by a slightly lesser amount than expected. This may be due to the difficulty of the sample discrimination task at large eccentricities, particularly for naive subjects (Supplementary Fig. 1e). Despite this, we see for both the model and the human subjects that the stimulus patch size at which sample and family discrimination performance is equal grows approximately in proportion to eccentricity. This is analogous to the behavior of visual crowding, for which the minimum distance at which targets can be recognized among distractors grows in proportion to eccentricity[13,14].

**Multiple factors underlie opposing effects.** Conceptually, the opposing effects of stimulus size on family and sample discrimination can be understood as arising from the degree of convergence of summary statistics. But their emergence from the specific computations of our model, which uses populations of localized overlapping receptive fields, is more nuanced, and worth dissecting. As an indication of this, note that no combination of pooling region size or physiologically plausible SNR yields chance-level sample discrimination performance for the largest stimuli (Fig. 3). This would seem contradictory to the conceptualization of these stimuli as visual metamers (physically different but perceptually indistinguishable)[18]. In particular, since the stimuli are synthesized so as to match the same statistics that

are measured by the observer model, and those statistics are exactly matched for the largest-diameter stimuli, why does sample discriminability not fall to chance at the largest stimulus sizes?

As in the human visual system, the observer model measures sensory information over localized regions. Information useful for sample discrimination is available when statistics have not converged within these localized regions, and thus differ for each sample. In the simulations, this occurs under two different conditions. First, if these regions are smaller than the image over which statistics were matched during stimulus generation, their responses will generally not be equal to those of the original texture. This is evident both in Fig. 1c and in our model simulations in Fig. 3b, where sample discrimination is always higher than family discrimination when pooling regions are small (e.g., when scaling is matched to V1 receptive field sizes). Second, even if pooling regions are very large, the stimulus is windowed by an aperture, and some pooling regions will necessarily overlap the aperture boundary, which will affect their responses[47,48]. Since these pooling regions summarize statistics only over the portions of the stimulus they cover, the statistics will again not be fully converged and their values will provide information useful for sample discrimination. To elucidate this issue, we simulated the discrimination performance of an observer model that utilizes responses from a single pooling region. We then examined performance as a function of the overlap between the pooling region and stimulus. Family discrimination is poor when ovlerap proportion is small, but improves with increasing overlap (Fig. 5a, red line). In contrast, this same variability in statistics leads to good sample discriminability when overlap is small, worsening as the stimulus overlap increases (Fig. 5a, black line).

Given the two effects described above, why does the full model exhibit opposing effects of stimulus size on sample and family discrimination? In particular, since the pooling regions (analogous to the receptive fields of the visual system) are of fixed size and location, and noise is independent across pooling regions, one might expect that as the stimulus increases in size and encroaches on more of them, each would provide additional information for both tasks. But two additional population-level factors cause the sample discrimination performance to fall relative to family discrimination performance: the decreasing influence of the aperture boundary with stimulus size, and response normalization.

The first factor is illustrated in cartoon form in Fig. 5. When the stimulus is small, only a small set of pooling regions overlap the stimulus and most only do so partially (Fig. 5b). Under these conditions, the model predicts sample discrimination performance should be much better than family discrimination performance. As the size of the stimulus increases, a larger proportion of receptive fields are fully covered by the stimulus (Fig. 5c). More specifically, because the area of the stimulus aperture grows faster than its circumference, the proportion of receptive fields that fall within the center (as opposed to straddling the border) increases (e.g., from 1/7 to 7/19 in the schematics of Fig. 5b and c). The increased proportion of fully overlapping receptive fields shifts the performance balance from sample toward family discrimination. A similar logic holds for receptive fields of the visual system, and provides an explanation for the failure to obtain full visual appearance matching when statistically matched stimuli are presented within small apertures[49]. In contrast, metamers indistinguishable to human observers can be achieved when samples are statistically matched within each of the V2-sized pooling regions covering a large stimulus, mitigating the effect of the stimulus aperture[18,50,51].

A final crucial factor contributing to opposing effects is the gain control (normalization) mechanism operating across pooling regions. The resulting surround suppression reduces the signal

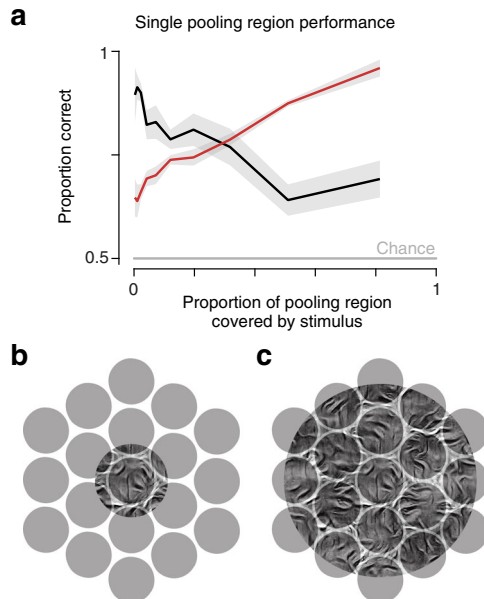

**Fig. 5 Model discrimination performance, computed for a single pooling region, varies depending on the portion covered by the stimulus.** **a** Average family (red) and sample (black) discrimination performance of single pooling regions as a function of the proportion covered by the stimulus. Simulations include pooling regions that are larger than the stimulus aperture, as well as those that are offset so as to extend beyond the stimulus border. Shaded region indicates mean ± SEM across performance over all geometries. **b** Schematic depiction of pooling region coverage for a small stimulus. **c** Schematic depiction of pooling region coverage for a large stimulus (note: example pooling regions in **b** and **c** are depicted in stylized form, as nonoverlapping circular discs—actual model pooling regions are smoothly overlapping and polar-separable with eccentricity-dependent sizes.) Source data are provided as a Source Data file.

strength, and thus the effective SNR, arising from each pooling region. Simulations of the observer model with normalization removed exhibit increasing performance for both family and sample discrimination with stimulus size (Supplementary Fig. 2). As the stimulus grows, it covers more pooling regions, each adding to the information that can be used for discrimination. For family discrimination, performance improvement is due both to this increase in information, as well as the convergence of the summary statistics within the larger proportion of pooling regions substantially covered by the stimulus. For sample discrimination, these two effects are opposed, although the former is dominant, and net performance still grows with stimulus size (albeit at a slower rate—see Supplementary Fig. 2b). Normalization reduces response strength as the stimulus grows, reducing performance in both tasks, to the extent that sample discrimination performance now falls with stimulus size. This result suggests that surround suppression induced by normalization, which has not been included in previous models of summary statistics[18,26,51,52], may play an essential role in limiting peripheral vision[42].

## Discussion

The nature, purpose, and capabilities of peripheral vision are fundamental aspects of vision science, but still hold many mysteries. Limitations of peripheral vision have been extensively documented[26], and a number of authors have suggested that these are complemented by benefits in coding efficiency[53], or the extraction of specific information that facilitates rapid assessment

of the whole visual field or scene[54–56]. The statistical summary hypothesis offers a specific instantiation of these ideas, positing that the visual system computes statistical summaries over spatial regions that increase in diameter with distance from the fovea, consistent with the approximately linear increase in receptive field diameter with eccentricity[9,11,15–18]. In this article, we have demonstrated a consequence of this hypothesis that directly reveals the tradeoff between the benefits and limitations of statistical summarization. When discriminating different types of texture, human observer performance increases with the size of the stimulus, but when discriminating different samples of the same type of texture, performance decreases. We have shown that a computational observer model that mimics basic physiological properties of neural populations in ventral stream areas V2 or V4 can predict both effects.

Our observer model is a generalization of one used previously to describe wide field-of-view images that are metamers, in that they are perceptually indistinguishable by human observers when matched for local texture statistics[18]. In particular, the current work augments that model to include gain control and noise, allowing it to characterize more nuanced discrimination performance (as opposed to only specifying perceptual equality of stimuli), and providing a consistent extension of the global model to locally windowed stimuli. Model simulations indicate that localized patches of texture samples from the same family are generally not metameric (i.e., discrimination based on model responses is above chance) regardless of window size (Fig. 3), primarily because pooling regions (physiologically, receptive fields) overlapping the boundary of the stimulus aperture provide information supporting sample discrimination. This is consistent with the perceptual data from our experiments (Figs. 2 and 4) as well as data from previous experiments[49]. Thus, contrary to[49], we conclude that the discriminability of local texture stimuli is consistent with the summary statistics hypothesis.

Our results are consistent with many of the well-documented limitations of peripheral vision, and offer an experimental method for their characterization. In particular, there is an extensive literature on visual crowding, in which target objects become difficult to recognize when surrounded by other distracting objects[11]. Recent studies have used a variety of methods and stimuli to demonstrate that many aspects of crowding can be explained by the compulsory pooling of features[15], or statistical summaries[17], computed within spatial regions that grow with eccentricity[18,26]. Our results generalize and strengthen the support for this hypothesis, using experimental stimuli and tasks that are distinct from those conventionally used for crowding. In particular, our stimuli are not defined in terms of explicit objects or features arranged in specific configurations, but are synthesized using a stochastic process that matches the responses of a neural population model. As a result, the stimuli are homogeneous, with their entire content providing information relevant for both tasks. In addition, our experimental tasks test discriminability (as opposed to recognition), allowing assessment of sensitivity to continuous sensory attributes rather than discrete classes or types. Finally, the use of two forms of discriminability exposes the benefits, as well as the limitations, of statistical summary.

A fixed pooling mechanism, as implemented in our model simulations, is arguably the most parsimonious explanation of the experimental results presented here. Previous work demonstrates that fixed pooling can also account for crowding and metamerism[17], even in the presence of attentional cueing[18], or other configural cues[52]. Nevertheless, recent work has suggested that fixed pooling may be insufficient to explain other crowding-related effects in peripheral vision[51,57–60]. Some authors have suggested that top-down feedback related to global scene configuration may be required to account for crowding and

metamerism[51,57–59], while others have suggested that nonlinear neural response properties might suffice to account for some results inconsistent with a fixed pooling model[42,60]. For example, surround suppression (the reduction of response that occurs when stimuli extend beyond the classical receptive field) has been extensively documented in V1[61–63] and extrastriate areas such as V2[31,64], and has been explicitly linked to performance degradation in population decoding of stimulus information[42]. Moreover, previous results demonstrate that changes in the relative gain of center and surround normalization mechanisms can mimic receptive field size changes[65], and that normalization strength can be gated by surrounding image statistics[66]. In our observer model, we found that the inclusion of surround suppression was critical in matching human performance.

Our experiments, and model, are built on the backbone of a particular statistical characterization for visual texture[23]. That model has proven useful: it is reasonably compact (~700 statistics), it is capable of capturing the appearance of a wide variety of different naturally occurring textures[23,24,49], and it preserves image attributes that drive V2 and V4 neurons robustly and selectively[27–32]. Nevertheless, it was developed by hand, using a qualitative and somewhat ad hoc procedure for selecting the set of features, which are surely are not perfectly aligned with the information captured in any particular neuronal population. This misalignment presumably underlies some of the quantitative differences between the opposing effects exhibited by our model and those of human subjects. Recent literature offers some hints as to what might be missing, or included unnecessarily. Perceptual studies of global stimuli suggest that the model is missing important features of natural images, especially those associated with extended contours or boundaries[50,51]. Consistent with this, physiological studies have found selectivity in V2 neurons to extended contours and multi-point correlations associated with corners and junctions[67,68]. Another study examined representation of periodic texture families using fMRI and EEG and found a set of statistics related to rotation invariance in human cortical area V3[69]. Although these studies provide intriguing suggestions for model enhancements, it is difficult to gather sufficient experimental measurements (either perceptual[25] or physiological[70]) to uniquely constrain a full model, given the high dimensionality of the space of visual images. An alternative is to use optimization methods, now common in machine learning, to train a model on large ensembles of photographic images[71–74]. Deep neural networks may prove useful in this regard, as they appear to learn complex features that bear some resemblance to late-stage visual representations[75,76] (some of which are texture-like[77]) and thus offer a potential source of summary statistics of relevance to human perception[78,79].

The perception of complex sensory patterns emerges from neuronal activity in a cascade of functional areas. The stimulus response properties of each successive area are constructed from those of the previous area, and these transformations appear to follow a canonical form[80–83], fusing specific combinations of afferents to generate new feature selectivity, and pooling over collections of these newly formed features to enable new invariances[29,43]. Although these systems also contain substantial feedback connections, the representation of immediate sensory information in any given area must come from earlier stages. If perception of a proximal stimulus is based only on information that survives this sequence of transformations, we expect that the loss of information that occurs in each area should engender opposing effects in the perceptual discriminability of the corresponding features. The current paper demonstrates this for visual features associated with areas V2[27] and V4[28] constructed by spatial averaging of particular combinations of V1 afferents[23]. It should be possible to generate such effects for features

represented in other visual areas, such as V1 or the retina, or later stages of the ventral or dorsal streams. Moreover, we would expect to see analogous effects in other sensory modalities. Indeed, previous studies have demonstrated opposing effects in the perception of sound textures of varying duration[22], and shown that these can be explained with an observer model based on temporal averaging of frequency-tuned peripheral afferents[36]. It seems likely that portions of the visual system would also exhibit temporal pooling, and perhaps pooling over other feature attributes (e.g., orientation). The opposing behaviors of family and sample discriminability provide a compelling signature of the corresponding tradeoff in selectivity and invariance of the underlying neural representation[29,43].

## Methods

**Stimulus generation.** We generated synthetic texture stimuli using the analysis–synthesis procedure described by Portilla and Simoncelli (2000)[23] (software and examples are available at www.cns.nyu.edu/~lcv/texture/). We measured the statistics of $320 \times 320$ pixel grayscale texture photographs, each of which served as the prototype for a texture "family." Each image was decomposed using a steerable pyramid[84], consisting of a bank of filters with four orientations and four spatial scales that tile the Fourier domain and constitute an invertible linear transform (technically, a tight frame). For each filter, we computed the linear responses as well as the local magnitude responses (square root of sum of squared responses of the filter and its Hilbert transform), in rough correspondence with responses of V1 simple and complex cells, respectively. We then computed pairwise products of responses at different positions (for each orientation and scale, within a $7 \times 7$ spatial neighborhood) for both sets of responses, and (for the magnitude responses only) across different orientations and scales. We also included products of linear filter responses with phase-doubled responses at the next coarsest scale. All of these pairwise products were averaged across the spatial extent of the image, yielding covariances. The covariances of the linear responses are second-order statistics, in that they represent averages of quadratic functions of pixel values. The covariances of magnitudes (and phase-doubled responses) are of higher order, due to the additional nonlinearities in the magnitude and phase-doubling computations. We additionally computed the average magnitude within each frequency band and the third- and fourth-order marginal pixel statistics (equivalently, the skew and kurtosis).

We generated $512 \times 512$ pixel synthetic textures for each family by initializing with an image of Gaussian white noise and adjusting it until it matched the model summary statistics computed on the corresponding original image[23]. Note that the global statistics of the original image are matched over the full synthetic image, in contrast with the synthesis of metamers in Freeman and Simoncelli (2011)[18] (in which statistics are matched over overlapping local windows covering the image), or the local stimulus patches in[49] (in which statistics are matched anew for each stimulus window size). For each texture family, we generated 15 different samples by initializing with different noise seeds. We chose four texture families from the set of 15 used in Freeman, Ziemba et al. (2013)[27], selecting those that were most difficult to discriminate so as to avoid ceiling effects in the current experiments. We used all four texture families for the simultaneously presented task, but restricted this to three families for the sequentially presented task, which has substantially longer trials (Supplementary Fig. 1a, c). To vary the stimulus size, we cropped square regions with widths 64, 128, 192, 320, 512 from the center of each synthesized texture (corresponding to 0.8, 1.6, 2.4, 4, 6.4 degrees of visual angle for our standard configuration). For presentation, we vignetted each square image with a circular aperture consisting of a raised cosine edge and a flat top which covered 7/8 of the width of the image.

To quantify the convergence of statistics with stimulus aperture size (Fig. 1c), we measured the statistics from subimages cropped from the full synthetic textures (that were generated so as to match all statistics over their full $512 \times 512$ pixel extent). We measured statistics from 15 samples across 15 different texture families (those used in Freeman, Ziemba et al. (2013)[27]). We cropped square regions from the center of these images with widths ranging from 64 pixels to 448 pixels in steps of 64 pixels. We computed the standard deviation of each individual statistic across 15 samples for each crop size and texture, and divided this value by the mean across samples to obtain the coefficient of variation. The median and interquartile range for each class of statistics are plotted in Fig. 1c.

## Psychophysics

*Observers and stimulus presentation.* Eleven human observers performed sample and family discrimination tasks. All observers had normal or corrected-to-normal vision. Four observers naive to the purpose of the studies performed experiment 1 (Fig. 2f; Supplementary Fig. 1b), Three naive observers and the first author performed experiment 2 (Fig. 2g; Supplementary Fig. 1d), and three naive observers and the first author performed experiment 3 (Fig. 4b; Supplementary Fig. 1e). Experimental procedures for human subjects were approved by the Institutional Review Board at New York University. All subjects provided written informed consent. Subjects sat in a darkened room 46 cm from a $41 \times 30$ cm flat CRT monitor. Their heads were

stabilized via a chin and forehead support. We presented stimuli using MATLAB (MathWorks) and MGL (available at http://justingardner.net/mgl/) on an Apple Macintosh computer. For all experiments, we presented images at a resolution of 80 pixels/degree when presented at 4 degrees eccentricity (as in Freeman, Ziemba et al. (2013)[27]). Stimuli presented at 8 and 16 degrees eccentricity were shown at 40 and 20 image pixels/degree, upsampling the original images by a factor of 2 and 4 in each direction. A 0.25 degree fixation square was shown throughout each trial, and subjects were instructed to fixate whenever it was on the screen.

*Tasks.* Subjects performed sample and family discrimination in two separate sessions on different days, with the order randomized across subjects. Each subject performed several dozen training trials with feedback on each task before starting the main experiment. Within each session, stimulus conditions were grouped into blocks of trials to facilitate performance. In addition, subjects completed a small number of practice trials with feedback at the beginning of each block to familiarize themselves with the block condition (4 practice trials in experiments 1 and 3, 6 practice trials in experiment 2). We excluded these practice trials from analysis, and no feedback was provided on analyzed trials. Grouped conditions were defined by size and eccentricity, as well as a particular texture family in the sample discrimination session and a particular texture family pair in the family discrimination session. In experiment 1, a condition block of 32 trials was shown once (Fig. 2f). In experiment 2, condition blocks were 18 trials and each block was shown to a subject two times for a total of 36 trials per condition (Fig. 2g). In experiment 3, a condition block of 40 trials was shown once (Fig. 4b). Trials within a block differed based on counterbalanced trial order, and the particular samples shown (randomly drawn from a set of 15 for each family). In all experiments, stimuli of different sizes were presented such that their inner edges (closest to the fixation point) were aligned. Such an organization ensures that larger images do not contain stimulus content closer to the center of gaze.

In the sample discrimination task, each trial made use of two synthetic texture samples drawn from the same family (one of them presented twice). In the family discrimination task, each trial made use of three synthetic texture samples, one drawn from one family and two drawn from another. In both discrimination tasks, subjects were instructed to identify the sample that was different (either in its detail, for the sample task, or in its category, for the family task). The sequential task used an AXB design. Subjects viewed three images presented for 100 ms in the same location separated by 900 ms. Subjects had unlimited time after each trial to respond with a button press, indicating whether the first or last image presented was different from the other two. The spatial task used an oddity design. On each trial, all three stimuli were presented simultaneously for 300 ms, at locations equidistant from fixation (one above, one to the lower left and one to the lower right). Subjects had unlimited time after each trial to indicate the location of the image that was different with a button press.

**Observer model predictions.** We constructed an observer model based on the analysis portion of the Freeman and Simoncelli model for full-field visual metamers[18] (software available at https://github.com/freeman-lab/metamers). The model computes summary statistics of an image within pooling regions whose size and number are fully determined by a positive real-valued scaling parameter (*s*). Specifically, all statistics are computed as spatially weighted averages within a set of *P* localized but partially overlapping windowing functions, which are designed to tile the image (i.e., their sum is 1). These functions are smooth, and separable with respect to polar angle and log eccentricity so that both their radial and angular extent grow in proportion to eccentricity. The aspect ratio of radial to circumferential width was ~2. For our analysis, we used an image region measuring 640 × 640 pixels, simulated to cover 8 × 8 degrees of visual angle. The fixation point of the model was placed outside the image region, at pixel coordinates (320.5, −256), so as not to simulate responses from many pooling regions that never covered the stimulus. We used all of the default parameters in specifying the model. Details of pooling region construction can be found in Freeman and Simoncelli (2011)[18].

Within each pooling region, the model extracts *N* = 668 spatially averaged summary statistics. These statistics include most of those used for stimulus generation: covariances across position for linear and magnitude responses (100 and 400 values, respectively), covariances across scale and orientation for magnitude responses (48 and 24 values), and covariances across scale for linear and phase-doubled linear responses (96 values), resulting in a total of 668 statistics per pooling region. These particular statistics have been shown to be most important for predicting perceptual[27] and neuronal[28,30] sensitivity to naturalistic image structure. We omitted marginal pixel statistics and the magnitude means of the original texture model[23].

Different groups of statistics span very different numerical ranges. To equalize the responses, we rescaled each statistic by its standard deviation over a large set of natural images. Specifically, we measured model responses across 3967 grayscale images from the Van Hateren database[85] using pooling scale factor *s* = 0.5. We computed the standard deviation of each statistic measured across the 777,532 different pooling regions in these data, then divided the value of each statistic in response to our experimental images by this standard deviation. We found that responses of this normalized model to our experimental stimuli also had an average standard deviation roughly equal to one. Observer model results are qualitatively similar when z-scoring across experimental stimuli.

After measuring these rescaled statistics from each experimental image, the statistics are then adaptively normalized by their Euclidean norm, $\left[\sum_{i=1}^{P}\sum_{j=1}^{N} r_{ij}^2\right]^{1/2}$, where the sum is over all *N* statistics in all *P* pooling regions, and then corrupted with additive Gaussian noise with standard deviation σ. The normalized statistic responses make up the response vector **R** for a particular image, which has the dimensions *P* × *N* (number of pooling regions by number of statistics). For a given trial, a vector of responses (overall statistics and pooling regions) are computed for each stimulus, $\mathbf{R}_A$, $\mathbf{R}_X$, and $\mathbf{R}_B$ (corresponding to the AXB task design). For the family discrimination task, images A and B are samples from two different families, and image X is a different sample from one of these families. Thus, $\mathbf{R}_X$ will differ from one of $\mathbf{R}_A$ or $\mathbf{R}_B$ due to sample variability, as well as the additive response noise. For the sample discrimination task, images A and B are two different samples from the same family, and image X is exactly the same as one of these. In this case, $\mathbf{R}_X$ will be identical to $\mathbf{R}_A$ or $\mathbf{R}_B$, except for the variations caused by the additive response noise.

To render the model's decision on each trial, we compute the Euclidean distance between responses to image X and those of each of the other two images:

$$D_A = \left\|\mathbf{R}_A - \mathbf{R}_X\right\|_2$$
$$D_B = \left\|\mathbf{R}_B - \mathbf{R}_X\right\|_2 \tag{1}$$

The model's decision corresponds to the maximum of these two values. Note that this simple distance comparison is not the statistically optimal strategy for an AXB-type decision task[86]. However, the optimal decision rule differs only slightly in performance. As such, we opted to use this simple read-out strategy of sensory differences for both tasks.

To estimate the proportion of correct model responses, we simulated 10,000 trials for each different size condition. Each trial was randomly drawn from four experimental conditions (Supplemenbtary Fig. 1c), and each image within a trial was randomly drawn from 15 samples of 4 families, mirroring the perceptual experiments. Model performance was qualitatively similar across the different sample and family discrimination tasks, as for human observers. Model performance is computed for three different scaling values, approximating receptive field scaling in visual areas V1 (*s* = 0.21), V2 (*s* = 0.46), and V4 (*s* = 0.84)[18]. To find a physiological plausible value for the noise level, σ, we first derived the signal-to-noise ratios from a previously collected dataset of single unit recordings from V1 and V2 of anesthetized macaque monkeys[27,29]. For each recorded neuron, we computed the variance in average firing rate elicited by 225 synthetic texture images (15 samples from 15 families) as a measure of signal strength. We computed the noise strength by measuring variance in response across 20 repeats of each image and averaged across all 225 images. We computed the SNR (ratio of the two values) for 102 V1 neurons (mean SNR = 0.64 ± 0.81) and 103 V2 neurons (mean SNR = 0.45 ± 0.39). We determined an analogous quantity for the model by computing the variance of each statistic across all experimental stimuli and pooling regions that covered the stimulus, and dividing by σ². Signal variance was somewhat different for each scaling, so we chose a σ for each *s* that yielded an average SNR of 0.5, approximately matching the physiological value (when *s* = 0.21, σ = 0.007 yields mean SNR = 0.5 ± 0.35; when *s* = 0.46, σ = 0.011 yields mean SNR = 0.5 ± 0.37; when *s* = 0.84, σ = 0.013 yields mean SNR = 0.5 ± 0.34). All model parameters were chosen and held fixed for all experimental conditions except for the scaling factor (*s*), and the noise level, (σ).

We performed model simulations with the normalization removed to illustrate its importance in matching human performance (Supplementary Fig. 2). All other procedures were the same as in the full model, except for the need to adjust σ to achieve the same SNR values as in Fig. 3b. (when *s* = 0.21, σ = 3.1 yields mean SNR = 0.5 ± 0.34; when *s* = 0.46, σ = 2.4 yields mean SNR = 0.5 ± 0.35; when *s* = 0.84, σ = 2.0 yields mean SNR = 0.5 ± 0.29).

Model performance for single pooling regions was computed in the same way as for the full model. For this analysis, we used pooling regions drawn from a model with *s* = 0.6, so as to be intermediate between V2 and V4 scaling values. We also simulated the same stimuli presented at 3 different eccentricities (4, 8, and 16 degrees) to increase the variation in pooling region size and stimulus overlap. We determined the proportion overlap by taking the inner product between the pooling window and stimulus aperture and dividing by the integral of the pooling region. We then averaged performance across pooling regions and stimuli at all sizes within 10 logarithmically spaced bins of overlap proportion.

**Reporting summary**. Further information on research design is available in the Nature Research Reporting Summary linked to this article.

## Data availability
The data that support the findings of this study are available in a public repository (OSF: https://doi.org/10.17605/OSF.IO/GFEPH)[87]. Source data are provided with this paper.

## Code availability
All code used for data analyses and computational simulations are available in a public repository (OSF: https://doi.org/10.17605/OSF.IO/GFEPH)[87].

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

## Acknowledgements

We thank Michael Schemitsch, Natalie Pawlak, and Rebecca Walton for their help in psychophysical data collection, and Mike Landy and Denis Pelli for useful discussions. This work was supported by National Institutes of Health (NIH) grant R01EY022428 and an HHMI investigatorship awarded to E.P.S., C.M.Z. was supported by NIH grants T32EY021462 and K99EY032102.

## Author contributions

C.M.Z. and E.P.S. conceived the study, developed the theory, and wrote the paper. C.M.Z. conducted the experiments, analyzed the data, and performed the simulations.

## Competing interests

The authors declare no competing interests.
