## [Peer Review File · Nature Communications]

REVIEWER COMMENTS

Reviewer #1 (Remarks to the Author):

In this work the authors test the hypothesis that peripheral vision relies on a summary statistics (SS) representation. This long-standing hypothesis has received much support in past work (duly cited) including by the authors. However most evidence supporting this hypothesis is in some way indirect, using stimuli and experimental designs that may not be ideally suited to probe SS representations, leading to considerable uncertainty. This manuscript presents what is, in our opinion, the most direct and convincing experimental test of the SS hypothesis for peripheral visual perception to date. Specifically, the authors use discrimination tasks with naturalistic synthetic textures. They derive the novel prediction that discrimination between textures with different SS (family discrimination) should improve with image size, whereas, somewhat surprisingly, discrimination between textures with the same SS but different pixel realizations (sample discrimination) should become worse for large images. The experiments provide strong and detailed support for the prediction. The manuscript is mostly clear and technically sound. We suggest below some points that could potentially be improved.

We enjoyed reading this manuscript, and believe this is a valuable and much needed contribution to the field.

NORMALIZATION: The authors mention that cross-receptive field normalization was required to match human performance (line 294). This seems an important methodological point that is not much discussed in the manuscript. How does the model output change without cross-receptive field normalization?

A related technical point is the choice of normalization for the summary statistics in the observer model. As described in Methods (line 379 and following), each statistic is normalized to a standard deviation of 1 in the Van Hateren database. But besides the scale of the standard deviation, the PS statistics can vary substantially in the magnitude of their mean values across natural images. It seems like not centering the statistics at 0 could lead a small subset of statistics to dominate the posterior normalization across all statistics described in the paragraph of line 384. What is the motivation for the particular choice used in the manuscript?

BOUNDARY EFFECTS: The section on line 191 and following emphasizes the relevance of the pooling regions at the boundaries of the stimulus for sample discrimination. The authors argue that this result can explain the failure of Wallis et.al. 2016 to find metamerism between images that are presented within small apertures. But in Wallis 2016, discrimination performance improves as the target size increases (their Figure 8) in a task analogous to the sample discrimination task described here. This would seem to run counter to the analysis made by the authors in this work. This seeming discrepancy with previous results could be acknowledged and discussed.

“COUNTERINTUITIVE” EFFECT OF IMAGE SIZE: Several parts of the text, including the abstract, emphasize that the reduced sample-discrimination performance for larger sizes is counterintuitive or paradoxical, and that one should expect performance to increase with stimulus size. Why one should expect this, however, is not entirely clear from the text. For instance, the introduction discusses the crowding literature, where spatial context impairs performance. Past work also showed that larger moving stimuli impair motion direction discrimination at high contrast (Tadin et al 2003 Nature). Line 87 and following make the point that “subjects could focus their attention on a small spatial portion of the larger stimuli, which would allow them to maintain high levels of performance *regardless of patch size*”, rather than improve performance for larger sizes. Perhaps the expectation of improved performance is related to neural noise. Implementing and simulating alternative models may be too much, but it would be useful to explain more clearly which models in the literature would predict that larger size equals better sample-discrimination performance.

COMPULSORY POOLING: A related point is whether the compulsory pooling, proposed here to explain the experimental observations (lines 94-97 and 270-273), is really compulsory, or whether instead it could be mitigated. For instance, could sample-discrimination performance at large sizes be

rescued by cueing a small portion of the larger images? What is the prediction of the model of fig 3a in that case? We're not suggesting that these experiments/simulations should necessarily be performed as part of this paper, but it seems a point worth considering, particularly in the discussion of the literature by the Herzog group, Wallis et al 2018, and our own (lines 293-295), given that the possibility of flexible pooling arising from segmentation or grouping cues is a major active area of peripheral vision research. A related, relevant reference is the recent review by Rosenholtz (JOV 2019) arguing in favor of the fixed windows pooling model.

Minor comments:

1) In the paragraph of line 27 the authors say "Here, we show that these peripheral losses are accompanied by a gain", but it is not clear exactly what this "gain" is, and how their experiments show this gain. Specifically, does the gain refer simply to the improvement of family-discrimination with image size? Or, for any given size, to the comparison against a model that does not use a summary statistics representation?

2) In line 75 the authors say that "The effect is consistent across stimuli and observers (Fig. 2f, red line)", but the referenced figure only shows mean performance across stimuli and observers, thus not really showing the mentioned consistency.

3) In the line of argument starting in line 228, it would seem that the authors suggest that the receptive fields straddling the boundary are required to solve the sample discrimination task. Is this correct? Should we expect performance to drop to chance if stimulus size is increased to infinity? If so, this could be explicitly stated. Nonetheless, it would seem like even without boundary straddling receptive fields there is variability in the fully-covered receptive fields that would allow to solve the task, making the straddling regions not strictly required to solve the sample discrimination task.

4) The size/scaling of the single-pooling-region model is not specified in the text.

5) Line 291 and following suggest deep NNs as a potential basis for probing SS representations of other visual features. The recent observation that deep NN feature activations are well described by elliptical distributions, and hence are well summarized by means and covariances at each layer (Vacher, Davila, Kohn, Coen-Cagli NeurIPS 2020), could be relevant from that perspective.

6) Typos:

line 33 "we show. through"

lines 39-40 open parenthesis

line 227. "overlapped receptive fields ..."

line 230. "(Fig. (Fig. 2f,3b)."

line 302 "...area should should engender..."

line 365 "Specifically, statistics are averages weighted a set of P..."

Reviewer #2 (Remarks to the Author):

First things first: Great paper; I have no serious concerns.

That being said, when I first read the manuscript (and skipped directly from the Introduction to the Methods, because that's the way I read), I was under the impression that the number of pooling regions (the variable P) was, in fact, a variable, which scaled with stimulus area. Since each of the N statistics within each pooling region is perturbed by (what is, I still assume to be an independent

sample of Gaussian) noise, that meant the Euclidean distance between response matrices would be expected to increase with stimulus area, and performance in the sample-discrimination task would have to fall.

I didn't find out that my impression was wrong until I got to Fig. 5. (So, here's one suggestion: when you introduce the variable P, make it clear that it doesn't vary with stimulus area.) Fig. 5 seems to suggest that all response matrices are the same size. I have to say that seems kind of strange to me. It seems as though you have effectively assumed that observers ignore noise that is generated in pooling regions that fall wholly outside only your largest stimuli, i.e. even on (blocked) trials that only contain small stimuli (and thus also contain wholly unstimulated pooling regions).

Regardless whether the inverse relationship between sample-discrimination accuracy with stimulus size is due to different samples looking more similar or identical samples looking more different, I don't see how this model could explain repulsion and assimilation associated with crowding in less naturalistic stimuli (e.g. doi:10.1167/10.8.13). Of course, I concede that no model can account for everything, and I am happy to concede those contextual effects as "beyond the scope" of the current manuscript.

Reviewer #3 (Remarks to the Author):

The manuscript by Ziemba & Simoncelli presents a set of fascinating findings to address the question of why our peripheral vision operates the way that it does. A great deal of recent work has examined the limitations of peripheral vision, including phenomena like crowding, as reviewed by the authors. Although suggestions have been made as to the potential usefulness of these operations, empirical demonstrations of these apparent benefits have proven elusive.

Following recent work suggesting that the periphery may represent the visual scene via summary statistics, here the authors demonstrate distinct patterns of performance for the discrimination of texture 'families' (whether synthetic texture images derive from a common image with matched statistics) vs. 'sample discrimination' (whether synthetic texture images are the very same image). As image size increases, sample discrimination performance declines, whereas family discrimination improves. The crossover between these abilities shifts to larger sizes as stimuli are presented further into the peripheral field. The authors replicate this with a texture-discrimination model that directly encodes summary statistics from these textures within localised receptive fields that increase in size with eccentricity. The implication is that our peripheral field, subserved by these large receptive fields, is optimised for discriminating between distinct texture patterns, at the expense of fine detail. Although the latter limitation has been extensively studied, the former benefits have received less demonstration. The argument here is compelling.

I very much enjoyed reading this manuscript. It is bristling with ideas and addresses a fundamental issue – why peripheral vision seems to limit information. Here we have a task where peripheral vision excels, showing a benefit for these operations. These kinds of demonstrations have been a kind of holy grail for many fields (e.g. the benefits of adaptation), so to see a task like this with clear data is fascinating. The behavioural experiments are carefully conducted and seamlessly integrated with computational modelling that together provides a range of insights. Both behavioural and modelling results are linked with prior findings in psychophysics and physiology to provide a firm foundation.

I do however have some issues with the manuscript as it stands, though these are mainly issues of clarification and points where conclusions are overstated.

1. The basis of sample vs family discrimination and their variation with image size

As the authors discuss, the finding that sample discrimination abilities (matching 2/3 images) decline with image size is somewhat counterintuitive. The manuscript mentions this, but focusses its explanation primarily on the family-discrimination task, which increases with size. I struggled to

understand the proposed basis for sample discrimination as a result and suggest that greater focus need be given here.

There seem to me to be two levels of explanation for these results. One relies solely on the statistics of the images themselves. As presented in Figure 1C, texture statistics are more variable between images when those images are small than when they are large. It makes sense then that sample discrimination should be successful with small images – there is a high likelihood that the oddball sample in the three-interval ABX task will be very different. This variability decreases with size (as Fig 1C shows), and so the statistics of the oddball are less likely to differ, making performance more difficult. For the family discrimination, the increased variability with small images is problematic since observers might erroneously take this difference as an indication of distinct family membership for the matched images (which match only in their broad statistics and are not the same image, as they are in the sample task). Is this a fair interpretation? If so then a greater link between the explanation and these image statistics in Fig 1C would help understand the basis for these abilities.

The second level to the interpretation here is the “observer model” that computes these statistics and uses them to make the same judgements, which performs similarly to the observers with the right parameters in place (particularly with respect to parameters for the scaling of receptive-field size with eccentricity and noise). The success of the model is used to argue for the “strong hypothesis” that observers perform this way because they have access only to this statistical information and not the fine details, as with the model. Confusingly however, the explanation in this section (p6) focusses on the effect of the overlap between the stimulus and receptive field boundaries. I find it hard to determine the model-based explanation of the sensitivity patterns as a result.

The description on lines 203-204 suggests to me that the model replicates the data for the same reason as above (the relationship between image size and variability in the images themselves). But this link could be made more explicitly. As it is, the explanation with partial overlap seems problematic in several ways.

First, this same process seems unlikely to be true for other visual judgements – with judgements of direction or orientation for instance, more information should lead to better discriminability. This is true for orientation judgements for instance (Mäkelä, Whitaker, & Rovamo, 1993), though there are of course exceptions e.g. with motion at high contrast when surround suppression becomes an issue (Tadin, Lappin, Gilroy, & Blake, 2003). What separates the current abilities is the nature of the stimuli themselves – as above, the statistical properties of the images become less variable as size increases. In other words, it seems to me that the explanation here needs both region of analysis (either image size or RF size), as is currently discussed, plus the statistical nature of these texture images. I do not think that partial overlap would be an issue for other visual judgements in the same way without this.

Second, the analysis of overlap is interesting but also confounds the size of the images – increasing size decreases their statistical variability, as above. Would the same result be found if an image of a fixed size were analysed by RFs of varying size (which would also alter ‘proportion covered of the pooling region’ as on the Fig 5a x-axis)? An analysis of this nature also seems closer to me to the situation in peripheral vision – given that images in the experiment expanded away from the fovea when their size increased, their representation also shifts towards neurons with larger receptive fields. This change in RF size with eccentricity also seems important to the success of the model – in Figure 3b, when shallow slopes for the change in RF size are used (akin to V1), model performance on the sample-discrimination task in particular lies flat at ceiling. Presumably this relates to the lack of change in size with eccentricity. Again this suggests to me that the issue is not one of partial overlap per se but rather a mismatch between the stimulus and RF size.

Along these lines, I note that there is also little consideration of alternative explanations for these patterns of performance. One relates to the above potential confound with eccentricity – that as size increases, so too does the eccentricity of the images. If the failure of sample-matching were not related to statistics but rather to the observer’s inability to compare features across the two images

(selected line elements or configurations of light/dark regions etc) then this would presumably decrease with eccentricity as acuity declined. Would the same result be obtained if patches were centred on the same eccentricity as they grew? The centroid location of elements has found to be most important to crowding effects (Levi & Carney, 2009); the same may be true here given that the centroid of these images shifts outwards with increasing size. Some consideration of this would also help to increase the generality of these findings.

Altogether, the manuscript would benefit from clarification on the proposed mechanisms underlying these abilities, a clearer distinction between the image-based and model-based levels of explanation, and some consideration of alternatives.

2. Image statistics and metamerism

The authors make a distinction between a strong variant of their hypothesis (that we ourselves perform these texture calculations and discard the fine detail), and a weaker version (that we perform some kind of texture calculation). The authors argue for the strong version on the basis of the match between the model and performance. But the circumstances here are rather limited for testing the model. In particular, I am sure that the model would do well to replicate crowding effects (where target recognition is disrupted by surrounding clutter), as with recent texture-based models of this process (Freeman & Simoncelli, 2011; Rosenholtz, Yu, & Keshvari, 2019), but I doubt that it could replicate performance without crowding. The striking thing about these effects is that observers can typically recognise an isolated letter or object with high accuracy, which drops markedly when surrounding flankers/clutter is added. Because these models discard positional information, they need various constraints in order to replicate this uncrowded baseline, e.g. that the output is restricted to specific locations (Balas, Nakano, & Rosenholtz, 2009). If we truly discard all fine details and have access only to statistical summaries, can we therefore read a single isolated letter in our peripheral vision?

Some caution is also required given that the experiments here compare only synthesised images against one another. Without testing a comparison to natural scenes, can the strong variant of the hypothesis really be asserted? This is discussed to some extent in the manuscript, with particular criticism reserved for a recent study (Wallis, Bethge, & Wichmann, 2016) which is argued to have reached “erroneous conclusions” about metamerism (line 258). Quite what the error is remains unclear to me however. Their argument is that observers can more readily discriminate natural scenes from texturised versions than they can discriminate two texturised images. I don’t see how the present results invalidate that finding given that natural scenes are not tested here. This seems to me a particular problem for the strong variant of the hypothesis.

The authors also discuss at several points the fact that their model never predicts chance-level performance, and argue that their participants behave similarly. This seems problematic to me given the fact that observers do fall to chance in this study – as in Figure 2 with the simultaneous matching task. It may be that the paragraph in the discussion on p7 is only referring to these model predictions and not to human data, but this could be more clear if so (on line 253 especially).

To be clear, I do think the data overall provide a compelling case for the authors arguments, but suggest that broader context be considered critically against these ideas.

3. Links to physiology

Claims are made at various points that the findings here directly relate to the physiology of the visual system, which at times are over-stated given the qualitative nature of these matches between data and model, the various assumptions required (e.g. averaging RF size across the population of neurons), and the relation to prior work. The results are described as indicating V2 and V4 neurons as a ‘likely candidate’ for these computations (line 156), which seems fair to me, but to then say directly that these findings provide direct evidence that these neurons represent these specific statistical representations of texture (line 158) seems a step too far. This is also true in the discussion where it is stated that “the current paper demonstrates this for visual areas V2 and V4” (line 303).

Although the similarities between the abilities observed herein and recordings from single neurons in these cortical regions is intriguing, it is problematic to then say that these behavioural and modelling results provide evidence for neural sensitivity in specific cortical regions. This is particularly so given that prior work (Freeman & Simoncelli, 2011) was most consistent with area V2, that others find scaling factors more consistent with area V1 (Wallis et al., 2019), and that there is evidence for the involvement of area V3 (Kohler et al., 2016). The locus of these texture processes therefore seems quite unclear to me, and although the results here can certainly constrain the potential processes performed by these neurons, they do not definitively determine the operation of specific neurons, as these statements suggest.

4. The nature of the texture images

It would be useful in evaluating the nature of “family discrimination” to see exactly the images that these samples derived from. The authors state that these images were specifically selected to be difficult to discriminate (line 331) in a prior study, suggesting that the precise images chosen are likely to matter. It is important therefore that we know what these were, ideally with the images themselves included in an appendix, for instance.

5. Methodological details

The methods report that 10 observers were tested, but each data figure reports only $n=4$. I assume the 10 are therefore spread over multiple experiments, but quite how this worked (and which were naïve) requires some explanation.

The sizes of stimuli are reported in pixels, but it would be useful to have their diameter in degrees (as plotted on Figure 4) for comparison to other visual processes.

Each condition was separated into distinct blocks of trials, the length of which is stated to be 20-40 trials at a time (line 346). Why the variation in range? Were some blocks combined?

In the model predictions section, the computation of image statistics involves a step in which the “response of each statistic” was divided by the standard deviation of these values. It’s not clear to me what the “response of each statistic” would represent here.

As above the arguments regarding the dependence of family vs sample discrimination on object size refer back to the variability in these statistics (Figure 1C). I cannot find however how these estimates of variability were calculated, nor the meaning of the “coefficient of variation” that is plotted. Some description in the methods would help.

6. Minor issues

The abstract describes the two tasks (family and sample discrimination) but without naming the latter in the brief methodological description. It would help to more quickly understand the results were this named prior to the description of the results.

The caption for Figure 3 describes the ‘observer’ as computing the Euclidean distance in statistical texture space (near the middle) which is somewhat confusing given that it refers to the ‘observer model’.

A few typos: ‘edge’ on line 66 is misspelt, as is ‘stimuli’ on line 193.

References

Balas, B., Nakano, L., & Rosenholtz, R. (2009). A summary-statistic representation in peripheral vision explains visual crowding. *Journal of Vision*, 9(12): 13, 1-18.

Freeman, J., & Simoncelli, E. P. (2011). Metamers of the ventral stream. *Nature Neuroscience*, 14(9), 1195-1201.

Kohler, P. J., Clarke, A., Yakovleva, A., Liu, Y., & Norcia, A. M. (2016). Representation of maximally regular textures in human visual cortex. *Journal of Neuroscience*, 36(3), 714-729.

Levi, D. M., & Carney, T. (2009). Crowding in peripheral vision: Why bigger is better. *Current Biology*, 19(23), 1988-1993.

Mäkelä, P., Whitaker, D., & Rovamo, J. (1993). Modelling of orientation discrimination across the visual field. *Vision Research*, 33(5-6), 723-730.

Rosenholtz, R., Yu, D., & Keshvari, S. (2019). Challenges to pooling models of crowding: Implications for visual mechanisms. *Journal of Vision*, 19(7), 1-25.

Tadin, D., Lappin, J. S., Gilroy, L. A., & Blake, R. (2003). Perceptual consequences of centre-surround antagonism in visual motion processing. *Nature*, 424(6946), 312-315.

Wallis, T. S. A., Bethge, M., & Wichmann, F. A. (2016). Testing models of peripheral encoding using metamerism in an oddity paradigm. *Journal of Vision*, 16(2), 4-4.

Wallis, T. S. A., Funke, C. M., Ecker, A. S., Gatys, L. A., Wichmann, F. A., & Bethge, M. (2019). Image content is more important than Bouma's Law for scene metamers. *eLife*, 8, e42512.

We thank all the reviewers for comments that have helped us to improve and refine our manuscript. Those sections with extensive revision have been highlighted in red, and we have added two supplementary figures, one breaking out detailed results by texture family and individual subject, and the other showing simulations from an observer model with normalization removed. Detailed responses to all comments are below.

REVIEWER #1

In this work the authors test the hypothesis that peripheral vision relies on a summary statistics (SS) representation. This long-standing hypothesis has received much support in past work (duly cited) including by the authors. However most evidence supporting this hypothesis is in some way indirect, using stimuli and experimental designs that may not be ideally suited to probe SS representations, leading to considerable uncertainty. This manuscript presents what is, in our opinion, the most direct and convincing experimental test of the SS hypothesis for peripheral visual perception to date. Specifically, the authors use discrimination tasks with naturalistic synthetic textures. They derive the novel prediction that discrimination between textures with different SS (family discrimination) should improve with image size, whereas, somewhat surprisingly, discrimination between textures with the same SS but different pixel realizations (sample discrimination) should become worse for large images. The experiments provide strong and detailed support for the prediction. The manuscript is mostly clear and technically sound. We suggest below some points that could potentially be improved.

We enjoyed reading this manuscript, and believe this is a valuable and much needed contribution to the field.

NORMALIZATION: The authors mention that cross-receptive field normalization was required to match human performance (line 294). This seems an important methodological point that is not much discussed in the manuscript. How does the model output change without cross-receptive field normalization?

Without normalization, discrimination performance generally increases with size in both tasks, but at different rates. This is expected: increasing stimulus size means that more receptive fields are driven, providing more information for either task. Normalization, a ubiquitous property of sensory neural responses, effectively reduces the gain, and thus the SNR, as the number of driven neurons increases. We have elaborated this in the manuscript, in both the Results and Methods sections, in addition to a supplementary figure showing model performance with normalization removed.

A related technical point is the choice of normalization for the summary statistics in the observer model. As described in Methods (line 379 and following), each statistic is normalized to a standard deviation of 1 in the Van Hateren database. But besides the scale of the standard deviation, the PS statistics can vary substantially in the magnitude of their mean values across natural images. It seems like not centering the statistics at 0 could lead a small subset of

statistics to dominate the posterior normalization across all statistics described in the paragraph of line 384. What is the motivation for the particular choice used in the manuscript?

Thanks for prompting us to clarify this. We now refer to this pre-processing step as “rescaling” (rather than “normalization”) in the methods to distinguish it from the physiologically-motivated cross-receptive field normalization. The rescaling of each statistic by its standard deviation over a large natural dataset is meant to reconcile the ranges of the different statistics. The mean and variance are highly correlated across the different statistics ($r = 0.97$), so the rescaling step brings both the mean and variance into similar ranges for each statistic. We have added language to the methods describing the motivations for our choice (and noting that z-scoring responses across experimental stimuli yields similar results).

*BOUNDARY EFFECTS: The section on line 191 and following emphasizes the relevance of the pooling regions at the boundaries of the stimulus for sample discrimination. The authors argue that this result can explain the failure of Wallis *et al.* 2016 to find metamerism between images that are presented within small apertures. But in Wallis 2016, discrimination performance improves as the target size increases (their Figure 8) in a task analogous to the sample discrimination task described here. This would seem to run counter to the analysis made by the authors in this work. This seeming discrepancy with previous results could be acknowledged and discussed.*

We appreciate you pointing this out. Comparing to the results of Wallis *et al.* 2016 is difficult, largely because of differences in stimulus generation. We synthesize large images with converged statistics, and the variability of statistics in our stimuli with stimulus size comes only from the windowing. The Wallis stimuli are synthesized independently for each window size, using statistics that are gathered (from natural images) *at that window size*, and thus vary much more substantially as size changes. As a result, their observers exhibit more variable behaviors: some show improvement in synthetic sample discrimination as window size increases from the smallest size, while others do not (see their Figure 8, first column, “synth vs synth” condition). More importantly, note that *all* of their subjects exhibit a decrease in sample discrimination performance at larger sizes, consistent with our results (again, see their Figure 8).

Ultimately, we thought it was not worth dragging readers through the technical differences between the papers. The important difference is not in methods or data, but interpretation: we have shown that the summary statistics hypothesis, and its instantiation in the full-field metamer model of Freeman & Simoncelli (2011), *is consistent* with the failure to achieve metamerism with localized texture patches seen in both data sets. Our model simulations, and accompanying explanation (in the renamed “Multiple factors underlie opposing effects” section) demonstrate why this is the case. We have refined our language in both the results section where the boundary effects are described and in the discussion.

“COUNTERINTUITIVE” EFFECT OF IMAGE SIZE: Several parts of the text, including the abstract, emphasize that the reduced sample-discrimination performance for larger sizes is counterintuitive or paradoxical, and that one should expect performance to increase with

*stimulus size. Why one should expect this, however, is not entirely clear from the text. For instance, the introduction discusses the crowding literature, where spatial context impairs performance. Past work also showed that larger moving stimuli impair motion direction discrimination at high contrast (Tadin et al 2003 Nature). Line 87 and following make the point that “subjects could focus their attention on a small spatial portion of the larger stimuli, which would allow them to maintain high levels of performance *regardless of patch size*”, rather than improve performance for larger sizes. Perhaps the expectation of improved performance is related to neural noise. Implementing and simulating alternative models may be too much, but it would be useful to explain more clearly which models in the literature would predict that larger size equals better sample-discrimination performance.*

Thank you for pointing this out. We intended to say that the result is “counterintuitive” because performance worsens as more stimulus information becomes available. An ideal observer would improve (and indeed, without normalization, our observer model does show improvement - this is now in a Supplementary figure). And even if humans are incapable of optimal spatial integration over the larger patches, one might expect that their performance would at least remain the same (eg., that they continue to integrate over a sub-region of the stimulus). Importantly, our task is quite different from those typically used in the crowding literature, because the surrounding stimulus information is relevant for the task, and not a masking or distracting “spatial context.” As such, we don’t see that the phenomenon of crowding could provide an explicit prediction of our results. We have updated the manuscript to be more explicit about this, see highlighted text in the Introduction and Results sections. Also, thank you for pointing us to Tadin et al, 2003, which we cite as providing a precedent for our results (albeit under quite different conditions).

COMPULSORY POOLING: A related point is whether the compulsory pooling, proposed here to explain the experimental observations (lines 94-97 and 270-273), is really compulsory, or whether instead it could be mitigated. For instance, could sample-discrimination performance at large sizes be rescued by cueing a small portion of the larger images? What is the prediction of the model of fig 3a in that case? We’re not suggesting that these experiments/simulations should necessarily be performed as part of this paper, but it seems a point worth considering, particularly in the discussion of the literature by the Herzog group, Wallis et al 2018, and our own (lines 293-295), given that the possibility of flexible pooling arising from segmentation or grouping cues is a major active area of peripheral vision research. A related, relevant reference is the recent review by Rosenholtz (JOV 2019) arguing in favor of the fixed windows pooling model.

Thank you for this suggestion. We’ve added a new (highlighted) paragraph to the Discussion on the issues surrounding the fixed pooling assumption.

Minor comments:

1) In the paragraph of line 27 the authors say “Here, we show that these peripheral losses are accompanied by a gain”, but it is not clear exactly what this “gain” is, and how their experiments

show this gain. Specifically, does the gain refer simply to the improvement of family-discrimination with image size? Or, for any given size, to the comparison against a model that does not use a summary statistics representation?

The “gain” is meant to refer to the improved family discrimination (i.e., improved sample invariance). We’ve refined the description.

2) In line 75 the authors say that “The effect is consistent across stimuli and observers (Fig. 2f, red line)”, but the referenced figure only shows mean performance across stimuli and observers, thus not really showing the mentioned consistency.

We have added a supplementary figure (Supp Fig. 1) breaking out performance across subjects and stimuli.

3) In the line of argument starting in line 228, it would seem that the authors suggest that the receptive fields straddling the boundary are required to solve the sample discrimination task. Is this correct? Should we expect performance to drop to chance if stimulus size is increased to infinity? If so, this could be explicitly stated. Nonetheless, it would seem like even without boundary straddling receptive fields there is variability in the fully-covered receptive fields that would allow to solve the task, making the straddling regions not strictly required to solve the sample discrimination task.

Your interpretation is correct, boundary straddling neurons are not *required* but they do provide additional information supporting discrimination in our task. We have revised this section to lay out the full set of factors that lead to the observed influence of window size on discriminability. These include: the degree of convergence of the statistics within pooling regions, the number of pooling regions that are covered, the surround suppression arising from divisive normalization, and the noise level.

4) The size/scaling of the single-pooling-region model is not specified in the text.

Thanks - we have added this to the methods section.

5) Line 291 and following suggest deep NNs as a potential basis for probing SS representations of other visual features. The recent observation that deep NN feature activations are well described by elliptical distributions, and hence are well summarized by means and covariances at each layer (Vacher, Davila, Kohn, Coen-Cagli NeurIPS 2020), could be relevant from that perspective.

Thank you for the suggestion, we have modified the sentence and added references.

6) Typos:

line 33 “we show. through”

lines 39-40 open parenthesis

line 227. "overlapped receptive fields ..."

line 230. "(Fig. (Fig. 2f,3b)."

line 302 "...area should should engender..."

line 365 "Specifically, statistics are averages weighted a set of P..."

Thanks - all fixed.

REVIEWER #2

First things first: Great paper; I have no serious concerns.

That being said, when I first read the manuscript (and skipped directly from the Introduction to the Methods, because that's the way I read), I was under the impression that the number of pooling regions (the variable P) was, in fact, a variable, which scaled with stimulus area. Since each of the N statistics within each pooling region is perturbed by (what is, I still assume to be an independent sample of Gaussian) noise, that meant the Euclidean distance between response matrices would be expected to increase with stimulus area, and performance in the sample-discrimination task would have to fall. I didn't find out that my impression was wrong until I got to Fig. 5. (So, here's one suggestion: when you introduce the variable P , make it clear that it doesn't vary with stimulus area.)

We apologize for the confusion. The first paragraph of the "Observer model predictions" section states that P is determined by the scaling factor s , and is independent of stimulus window size. Note that this means some pooling regions are empty (for small stimuli). We have added a sentence near the end of the methods section making clear that all model parameters are held fixed for all experimental conditions except for the scaling factor (s), and the noise level (σ).

Regarding the conceptual aspect of the question: Effectively, P does increase with stimulus area (because pooling regions are fixed, and a larger stimulus will cover more of them, proportional to area), but this has little effect on performance. Our observer model uses relative Euclidean distance between stimuli of the same size, and so is unaffected by the dependence of Euclidean distance on stimulus area.

Fig. 5 seems to suggest that all response matrices are the same size. I have to say that seems kind of strange to me. It seems as though you have effectively assumed that observers ignore noise that is generated in pooling regions that fall wholly outside only your largest stimuli, i.e. even on (blocked) trials that only contain small stimuli (and thus also contain wholly unstimulated pooling regions).

We thought the simplest approach was to use the same of pooling regions for all conditions, regardless of whether they are stimulated or not (since presumably this must also be true of the visual system). So the pooling regions beyond the stimulus window are included in the observer model, but have minimal impact on average performance. We have run simulations using only those pooling regions that are significantly overlapped with the stimulus, and results were nearly identical.

Regardless whether the inverse relationship between sample-discrimination accuracy with stimulus size is due to different samples looking more similar or identical samples looking more different, I don't see how this model could explain repulsion and assimilation associated with crowding in less naturalistic stimuli (e.g. doi:10.1167/10.8.13). Of course, I concede that no

model can account for everything, and I am happy to concede those contextual effects as "beyond the scope" of the current manuscript.

Indeed we do not immediately see how our results and model can explain many phenomena in the literature associated with crowding, such as the Mareschal et. al. 2010 paper. Specifically, the model as currently implemented makes no distinction between target stimuli and task irrelevant distractors, and does not read out a stimulus estimate like an orientation judgment. Adding such features to this model is indeed "beyond the scope" here but could be an interesting future line of research.

REVIEWER #3

The manuscript by Ziemba & Simoncelli presents a set of fascinating findings to address the question of why our peripheral vision operates the way that it does. A great deal of recent work has examined the limitations of peripheral vision, including phenomena like crowding, as reviewed by the authors. Although suggestions have been made as to the potential usefulness of these operations, empirical demonstrations of these apparent benefits have proven elusive.

Following recent work suggesting that the periphery may represent the visual scene via summary statistics, here the authors demonstrate distinct patterns of performance for the discrimination of texture 'families' (whether synthetic texture images derive from a common image with matched statistics) vs. 'sample discrimination' (whether synthetic texture images are the very same image). As image size increases, sample discrimination performance declines, whereas family discrimination improves. The crossover between these abilities shifts to larger sizes as stimuli are presented further into the peripheral field. The authors replicate this with a texture-discrimination model that directly encodes summary statistics from these textures within localised receptive fields that increase in size with eccentricity. The implication is that our peripheral field, subserved by these large receptive fields, is optimised for discriminating between distinct texture patterns, at the expense of fine detail. Although the latter limitation has been extensively studied, the former benefits have received less demonstration. The argument here is compelling.

I very much enjoyed reading this manuscript. It is bristling with ideas and addresses a fundamental issue – why peripheral vision seems to limit information. Here we have a task where peripheral vision excels, showing a benefit for these operations. These kinds of demonstrations have been a kind of holy grail for many fields (e.g. the benefits of adaptation), so to see a task like this with clear data is fascinating. The behavioural experiments are carefully conducted and seamlessly integrated with computational modelling that together provides a range of insights. Both behavioural and modelling results are linked with prior findings in psychophysics and physiology to provide a firm foundation.

I do however have some issues with the manuscript as it stands, though these are mainly issues of clarification and points where conclusions are overstated.

1. The basis of sample vs family discrimination and their variation with image size

As the authors discuss, the finding that sample discrimination abilities (matching 2/3 images) decline with image size is somewhat counterintuitive. The manuscript mentions this, but focusses its explanation primarily on the family-discrimination task, which increases with size. I struggled to understand the proposed basis for sample discrimination as a result and suggest that greater focus need be given here.

There seem to me to be two levels of explanation for these results. One relies solely on the statistics of the images themselves. As presented in Figure 1C, texture statistics are more

variable between images when those images are small than when they are large. It makes sense then that sample discrimination should be successful with small images – there is a high likelihood that the oddball sample in the three-interval ABX task will be very different. This variability decreases with size (as Fig 1C shows), and so the statistics of the oddball are less likely to differ, making performance more difficult. For the family discrimination, the increased variability with small images is problematic since observers might erroneously take this difference as an indication of distinct family membership for the matched images (which match only in their broad statistics and are not the same image, as they are in the sample task). Is this a fair interpretation? If so then a greater link between the explanation and these image statistics in Fig 1C would help understand the basis for these abilities.

This is a nice statement of the overall effect, and is the primary conceptual message of our paper. But there is a subtle but important aspect missing from this explanation, that arises from a more mechanistic level of explanation. Your statement assumes that human performance reflects statistics computed across the *entire* stimulus. As we elaborate in the renamed section “Multiple factors underlie opposing effects” and in the Discussion, our observer model computes statistics within fixed pooling regions (corresponding to receptive fields of neurons in some visual area). This means that the statistics may (or may not) have reached convergence, depending on pooling region size and how much of the pooling region overlaps the stimulus. These factors do contribute to the discrimination performance, as shown in Results. We thought it important to spell out all of the factors in our model (and, we believe, in the human observers) that lead to the net performance. We have modified sections of the results to try to better explain our thinking.

The second level to the interpretation here is the “observer model” that computes these statistics and uses them to make the same judgements, which performs similarly to the observers with the right parameters in place (particularly with respect to parameters for the scaling of receptive-field size with eccentricity and noise). The success of the model is used to argue for the “strong hypothesis” that observers perform this way because they have access only to this statistical information and not the fine details, as with the model. Confusingly however, the explanation in this section (p6) focusses on the effect of the overlap between the stimulus and receptive field boundaries. I find it hard to determine the model-based explanation of the sensitivity patterns as a result.

The description on lines 203-204 suggests to me that the model replicates the data for the same reason as above (the relationship between image size and variability in the images themselves). But this link could be made more explicitly.

These comments were very helpful and we have attempted to clarify our reasoning throughout the manuscript to make clear which aspects of the model are necessary to reproduce performance. We have modified the observer model section of the Results, as well as completely rewritten/added a new section “Multiple factors underlie opposing effects” to try to better explain what we consider the key components of the model that allow it to replicate the

data. A major reason the model works stems from the explanation you gave above: the increasing convergence of the statistics with image size. However, the interaction between this effect and pooling regions of fixed size is somewhat complex, and this is what we explore with this second level of interpretation. Indeed, we find that gain control, in addition to pooling, is required for the full model to replicate human perceptual behavior.

As it is, the explanation with partial overlap seems problematic in several ways.

First, this same process seems unlikely to be true for other visual judgements – with judgements of direction or orientation for instance, more information should lead to better discriminability. This is true for orientation judgements for instance (Mäkelä, Whitaker, & Rovamo, 1993), though there are of course exceptions e.g. with motion at high contrast when surround suppression becomes an issue (Tadin, Lappin, Gilroy, & Blake, 2003). What separates the current abilities is the nature of the stimuli themselves – as above, the statistical properties of the images become less variable as size increases. In other words, it seems to me that the explanation here needs both region of analysis (either image size or RF size), as is currently discussed, plus the statistical nature of these texture images. I do not think that partial overlap would be an issue for other visual judgements in the same way without this.

These are good points, and we are mostly in agreement with your interpretation. We believe the core mechanisms of pooling and normalization are likely to affect many visual judgments. In many cases increasing stimulus size or information will not decrease discriminability, as you point out for orientation discrimination. We suggest this may be because with some stimuli, pooling mechanisms will not lead to the loss of much information. However, the statistical properties of the texture images we use here are to some extent general properties of naturalistic images. Smaller samples of sensory evidence will lead to more variability in summary statistics, while larger samples will lead to more stable statistics. So we would argue the opposing effects we demonstrate here may represent a general phenomena for discrimination of many complex, naturalistic stimuli.

However, we do think that aperture effects could contribute to other visual phenomena, as has been pointed out in the context of decoding orientation tuning (Carlson, 2014; Roth, Heeger, Merriam, 2018).

Second, the analysis of overlap is interesting but also confounds the size of the images – increasing size decreases their statistical variability, as above. Would the same result be found if an image of a fixed size were analysed by RFs of varying size (which would also alter 'proportion covered of the pooling region' as on the Fig 5a x-axis)? An analysis of this nature also seems closer to me to the situation in peripheral vision – given that images in the experiment expanded away from the fovea when their size increased, their representation also shifts towards neurons with larger receptive fields.

Yes, in fact this is partly the analysis we perform. The RFs were not all the same size (although they are depicted schematically that way in Fig. 5bc - we have added a note to the caption

emphasizing that the diagram is idealized). We used our observer model with eccentricity-dependent pooling region size to simulate this analysis, meaning pooling regions of multiple different sizes are included. Additionally we presented the same stimuli at multiple eccentricities to increase variation in pooling region size relative to the stimulus. This is now noted in the methods.

The same result of opposing effects of pooling region overlap holds for stimuli of a fixed size. However, given our analysis, the trend is somewhat noisier given the limited number of conditions and pooling regions. We averaged across different stimulus sizes to make the opposing effects clearer, and don't believe this detracts from the broader point.

This change in RF size with eccentricity also seems important to the success of the model – in Figure 3b, when shallow slopes for the change in RF size are used (akin to V1), model performance on the sample-discrimination task in particular lies flat at ceiling. Presumably this relates to the lack of change in size with eccentricity. Again this suggests to me that the issue is not one of partial overlap per se but rather a mismatch between the stimulus and RF size.

In fact, the V1 model shows opposing effects as well, and the sample discrimination performance appearing “flat” is, as you say, purely a product of being at ceiling because of SNR. You can see at an SNR of 0.125 that in fact performance is decreasing with size for the sample discrimination task, and decreases more dramatically for even lower SNR. We have added a sentence about ceiling effects in results.

But your larger point is correct: the reason V1 is not a good match to the human perceptual data is because the receptive fields are too small relative to the pooling regions that humans appear to be using. Even when a V1 sized pooling region is fully covered with our stimuli, it is taking a relatively small sample of the stimulus, yielding strongly variable responses across samples within a pooling region which provide too much sample discrimination information relative to family discrimination order to match the human data. We have added this explanation to the Results section.

Along these lines, I note that there is also little consideration of alternative explanations for these patterns of performance. One relates to the above potential confound with eccentricity – that as size increases, so too does the eccentricity of the images. If the failure of sample-matching were not related to statistics but rather to the observer's inability to compare features across the two images (selected line elements or configurations of light/dark regions etc) then this would presumably decrease with eccentricity as acuity declined. Would the same result be obtained if patches were centred on the same eccentricity as they grew? The centroid location of elements has found to be most important to crowding effects (Levi & Carney, 2009); the same may be true here given that the centroid of these images shifts outwards with increasing size. Some consideration of this would also help to increase the generality of these findings.

We do not consider what you describe an experimental confound.

(1) The distinction between computing statistical summaries and discarding the details and an “observer’s inability to compare features across images” is not clear to us. Subjects clearly have access to the statistics (as demonstrated by their family discrimination performance), but lose access to details allowing them to perform the sample discrimination well. We only add information at further eccentricities, so this extra information interfering with stimulus information at the nearest eccentricity is a striking effect that we do not believe can be explained by acuity.

(2) We believe the same result would be obtained if patches were centered at the same eccentricity as they grew. We did not do this because increasing size from the center moves relevant stimulus information closer to the fovea, eventually extending beyond it, and making it difficult to quantify effects of eccentricity. To lend credence to this, we can see evidence in the psychometric performance results of Wallis et al. (2016). Their Figure 8, left column shows 4 observers performing a sample discrimination task between synthetic images where images grow in size centered on the same eccentricity. While their stimulus construction differs from what we do here (see above reply to Rev 1), all observers exhibit performance decrements when the stimulus size increases beyond a diameter of roughly 4 degrees.

(3) We find the connection to findings from crowding paradigms such as Levi & Carney, (2009) somewhat difficult to reason about with respect to our results without actually performing a simulation. The fact that centroids of elements are important when relevant and irrelevant stimulus content is adjacent and interacting does not suggest an obvious connection to our task where the entire stimulus is a task relevant target. We absolutely agree these questions are crucial and interesting, but consider them to be in the “future work” category. We have modified portions of the Discussion to express this.

Altogether, the manuscript would benefit from clarification on the proposed mechanisms underlying these abilities, a clearer distinction between the image-based and model-based levels of explanation, and some consideration of alternatives.

These comments were immensely helpful - we have revised our manuscript so as to explain and distinguish these different issues, and hope you will find it improved.

2. Image statistics and metamerism

The authors make a distinction between a strong variant of their hypothesis (that we ourselves perform these texture calculations and discard the fine detail), and a weaker version (that we perform some kind of texture calculation). The authors argue for the strong version on the basis of the match between the model and performance. But the circumstances here are rather limited for testing the model. In particular, I am sure that the model would do well to replicate crowding effects (where target recognition is disrupted by surrounding clutter), as with recent texture-based models of this process (Freeman & Simoncelli, 2011; Rosenholtz, Yu, & Keshvari, 2019), but I doubt that it could replicate performance without crowding. The striking thing about

these effects is that observers can typically recognise an isolated letter or object with high accuracy, which drops markedly when surrounding flankers/clutter is added. Because these models discard positional information, they need various constraints in order to replicate this uncrowded baseline, e.g. that the output is restricted to specific locations (Balas, Nakano, & Rosenholtz, 2009). If we truly discard all fine details and have access only to statistical summaries, can we therefore read a single isolated letter in our peripheral vision?

It is an interesting question. The Freeman & Simoncelli 2011 and Rosenholtz et al 2019 simulations used the same basic observer model to demonstrate that the statistics are sufficient to synthesize recognizable isolated letters, implying that the information is retained. The underlying reason is closely related to our analysis of the boundaries of our stimuli, which provide information for sample discrimination. An isolated letter, which would be “seen” by many pooling regions, would be recognizable from the combined statistics of those pooling regions. This concept also underlies the substantial improvements in object recognition performance of artificial neural networks in recent years. The convolution and pooling architecture of those systems discards information through summarization, which provides invariances over spatial location and size (amongst other things) while preserving selectivity for object identity.

Some caution is also required given that the experiments here compare only synthesised images against one another. Without testing a comparison to natural scenes, can the strong variant of the hypothesis really be asserted? This is discussed to some extent in the manuscript, with particular criticism reserved for a recent study (Wallis, Bethge, & Wichmann, 2016) which is argued to have reached “erroneous conclusions” about metamerism (line 258). Quite what the error is remains unclear to me however. Their argument is that observers can more readily discriminate natural scenes from texturised versions than they can discriminate two texturised images. I don’t see how the present results invalidate that finding given that natural scenes are not tested here. This seems to me a particular problem for the strong variant of the hypothesis.

As we acknowledge in our response to Reviewer 1, we don’t take issue with the experimental results in Wallis et al. (2016), but we do disagree with some of the interpretation. In particular, we argue that the failure to achieve metamerism with windowed homogeneous texture patches is expected, and is completely consistent with a summary statistics pooling model of peripheral vision. We have clarified these arguments in the rewritten discussion.

That said, we do find their results comparing natural to synthetic images interesting, and view them as clear indicators that our current statistical model is missing features that are perceptually relevant. We believe this is also true physiologically: the texture properties captured by our model do drive V2 neurons well, on average, but the effect is weak in some cells. We’ve elaborated on this in the Discussion.

It’s worth noting that, unlike the global metamer paradigm of Freeman & Simoncelli 2011, extending our patch discrimination experiment to include natural photographs is not

straightforward. We use synthetic stimuli designed to be spatially homogeneous, and we rely on this property to interpret the results. In particular, the opposing effects can be understood to arise from a common set of statistics that underlie both tasks. Running the same experiment with inhomogeneous natural image patches would require careful re-evaluation of procedures for synthesizing samples, as well as more complex interpretive reasoning. We feel our current results stand on their own, and benefit from the focus of the experimental design.

The authors also discuss at several points the fact that their model never predicts chance-level performance, and argue that their participants behave similarly. This seems problematic to me given the fact that observers do fall to chance in this study – as in Figure 2 with the simultaneous matching task. It may be that the paragraph in the discussion on p7 is only referring to these model predictions and not to human data, but this could be more clear if so (on line 253 especially).

These are good points which we have attempted to clarify in the manuscript. We believe the simultaneous task presents extra challenges for discrimination that drive performance close to chance, and intended to mainly refer to model performance.

To be clear, I do think the data overall provide a compelling case for the authors arguments, but suggest that broader context be considered critically against these ideas.

3. Links to physiology

Claims are made at various points that the findings here directly relate to the physiology of the visual system, which at times are over-stated given the qualitative nature of these matches between data and model, the various assumptions required (e.g. averaging RF size across the population of neurons), and the relation to prior work. The results are described as indicating V2 and V4 neurons as a ‘likely candidate’ for these computations (line 156), which seems fair to me, but to then say directly that these findings provide direct evidence that these neurons represent these specific statistical representations of texture (line 158) seems a step too far. This is also true in the discussion where it is stated that “the current paper demonstrates this for visual areas V2 and V4” (line 303).

We have softened the wording of our claims, but we do think that the connection to a plausible underlying implementation is a strength of our approach. As such, we think it’s important to describe recent findings relating these particular statistics to physiological responses (there have been over half a dozen physiological studies).

Although the similarities between the abilities observed herein and recordings from single neurons in these cortical regions is intriguing, it is problematic to then say that these behavioural and modelling results provide evidence for neural sensitivity in specific cortical regions. This is particularly so given that prior work (Freeman & Simoncelli, 2011) was most consistent with area

V2, that others find scaling factors more consistent with area V1 (Wallis et al., 2019), and that there is evidence for the involvement of area V3 (Kohler et al., 2016). The locus of these texture processes therefore seems quite unclear to me, and although the results here can certainly constrain the potential processes performed by these neurons, they do not definitively determine the operation of specific neurons, as these statements suggest.

The current paper does not attempt to provide a precise statement regarding physiological locus for the opposing effects (we only say that they are qualitatively consistent with V2 and/or V4). But more generally, we do agree that perceptual scaling estimates do not provide definitive evidence for physiological locus of representation. We think the primary impediment stems from the difficulty of correctly and completely defining the relevant feature set from which statistics are computed. Specifically:

(1) The association of particular texture statistics with area V2 in Freeman & Simoncelli (2011) led directly to a physiological followup study (Freeman, Ziemba et al. 2013) that confirmed that V2 (but not V1) was sensitive to those particular statistics. This explicit confirmation demonstrates that these connections can be made given the right stimuli, experimental measurements, and interpretive logic.

(2) In Wallis et al. (2019), scaling appeared to be closer to V1 when comparing synthetic images with natural ones that had extensive or isolated features, such as long contours. We found this interesting, and interpret it as an indication that some perceptually-relevant features (e.g., combinations of collinear V1 responses) have been left out of the synthesis model. These missing features are only captured when the scaling parameter is set smaller than that required for the other features, and as a result, the scaling parameter used to fit the perceptual data is an under-estimate. This is a somewhat intricate argument, and tangential to the current paper, so we allude to it only indirectly in the Discussion paragraph about the choice of statistical features.

(3) Kohler et al. (2016) uses a texture model with more complex parameters that capture visual symmetries not constrained by the Portilla-Simoncelli statistics. As the authors note, it is fascinating (but unsurprising), that fMRI responses implicate areas beyond V2 in the representation of these particular statistics.

4. The nature of the texture images

It would be useful in evaluating the nature of “family discrimination” to see exactly the images that these samples derived from. The authors state that these images were specifically selected to be difficult to discriminate (line 331) in a prior study, suggesting that the precise images chosen are likely to matter. It is important therefore that we know what these were, ideally with the images themselves included in an appendix, for instance.

Thank you for suggesting this - we agree, and have added a supplementary figure showing an example of all of the texture families that samples are drawn from for each task. We also show

how different sample or family discrimination stimuli are easier or harder to discriminate. But note that all cases clearly exhibit the opposing effects of decreasing sample performance and increasing family performance with patch size. We chose families that were difficult to discriminate so as to avoid ceiling effects for moderate to large stimuli - most of the texture families we've worked with are relatively easy to discriminate. We have noted this in the methods to clarify our choices.

5. Methodological details

The methods report that 10 observers were tested, but each data figure reports only n=4. I assume the 10 are therefore spread over multiple experiments, but quite how this worked (and which were naïve) requires some explanation.

We agree this should have been better presented, and have clarified the details of this methods section. Additionally, we now show the individual performance of every subject in the new supplementary figure 1.

The sizes of stimuli are reported in pixels, but it would be useful to have their diameter in degrees (as plotted on Figure 4) for comparison to other visual processes.

We have added the diameter in degrees used in our standard stimulus configuration to the stimulus generation section, and added a sentence in the following section and paragraph to make clear that the same stimuli were shown at different sizes when presented further out in the periphery.

Each condition was separated into distinct blocks of trials, the length of which is stated to be 20-40 trials at a time (line 346). Why the variation in range? Were some blocks combined?

The blocking was different for each of the 3 main experiments, and this has now been explained in this section.

In the model predictions section, the computation of image statistics involves a step in which the "response of each statistic" was divided by the standard deviation of these values. It's not clear to me what the "response of each statistic" would represent here.

We attempt to consistently refer to each individual type of value computed by the model as a "statistic" or "summary statistic," to the value measured from an individual image as the "response" of that statistic, and to the value of all statistics in the model as the "response" of the entire model to that image. Perhaps in this sentence this is not fully clear so we have modified it to read "value of each statistic in response to our experimental images."

As above the arguments regarding the dependence of family vs sample discrimination on object size refer back to the variability in these statistics (Figure 1C). I cannot find however how these

estimates of variability were calculated, nor the meaning of the “coefficient of variation” that is plotted. Some description in the methods would help.

Thank you for pointing this out. We have added a description to the methods and explain the coefficient of variation in the figure caption.

6. Minor issues

The abstract describes the two tasks (family and sample discrimination) but without naming the latter in the brief methodological description. It would help to more quickly understand the results were this named prior to the description of the results.

We now specifically refer to samples as well as families when describing the tasks.

The caption for Figure 3 describes the ‘observer’ as computing the Euclidean distance in statistical texture space (near the middle) which is somewhat confusing given that it refers to the ‘observer model’.

We modified this sentence to read “The model computes...”

A few typos: ‘edge’ on line 66 is misspelt, as is ‘stimuli’ on line 193.

Fixed.

References

Balas, B., Nakano, L., & Rosenholtz, R. (2009). A summary-statistic representation in peripheral vision explains visual crowding. Journal of Vision, 9(12): 13, 1-18.

Freeman, J., & Simoncelli, E. P. (2011). Metamers of the ventral stream. Nature Neuroscience, 14(9), 1195-1201.

Kohler, P. J., Clarke, A., Yakovleva, A., Liu, Y., & Norcia, A. M. (2016). Representation of maximally regular textures in human visual cortex. Journal of Neuroscience, 36(3), 714-729.

Levi, D. M., & Carney, T. (2009). Crowding in peripheral vision: Why bigger is better. Current Biology, 19(23), 1988-1993.

Mäkelä, P., Whitaker, D., & Rovamo, J. (1993). Modelling of orientation discrimination across the visual field. Vision Research, 33(5-6), 723-730.

Rosenholtz, R., Yu, D., & Keshvari, S. (2019). Challenges to pooling models of crowding: Implications for visual mechanisms. Journal of Vision, 19(7), 1-25.

Tadin, D., Lappin, J. S., Gilroy, L. A., & Blake, R. (2003). Perceptual consequences of centre-surround antagonism in visual motion processing. Nature, 424(6946), 312-315.

Wallis, T. S. A., Bethge, M., & Wichmann, F. A. (2016). Testing models of peripheral encoding using metamerism in an oddity paradigm. Journal of Vision, 16(2), 4-4.

Wallis, T. S. A., Funke, C. M., Ecker, A. S., Gatys, L. A., Wichmann, F. A., & Bethge, M. (2019). Image content is more important than Bouma's Law for scene metamers. eLife, 8, e42512.

REVIEWER COMMENTS

Reviewer #1 (Remarks to the Author):

The authors addressed all our concerns to satisfaction. We find the new manuscript improved, in particular the new simulations and discussion of the requirement of normalization and surround suppression to observe the effects.

Reviewer #2 (Remarks to the Author):

this is a very nice paper

Reviewer #3 (Remarks to the Author):

The revised manuscript by Ziemba & Simoncelli presents a set of improvements to what was already a fascinating manuscript. The clarifications on model details have greatly improved the clarity of its operation, the new supplementary figures add details to help better understand both the data and the model, and revised text clarifies the position of these findings in the broader literature. I have no further major issues with the manuscript as it stands, though there are a few very minor details the authors may wish to clarify.

In the response to reviews the authors say that “As such, we don’t see that the phenomenon of crowding could provide an explicit prediction of our results”. However, line 169 of the revised manuscript states that “...populations of neurons in areas V2 and V4 are likely candidates for the locus of compulsory pooling giving rise to the opposing effects of stimulus size”. I take it that compulsory pooling is an apt description of the model processes here, but given the wide usage of this term to refer to models of crowding (following Parkes et al, 2001), a slightly different term would be ideal here, unless that equivalence is the authors’ intention.

Regarding the relation to Wallis et al (2019), the authors’ response states that “This is a somewhat intricate argument, and tangential to the current paper, so we allude to it only indirectly in the Discussion paragraph about the choice of statistical features”. This is generally fine, though the phrasing in the discussion is a little too indirect to be clear. Line 243 states “In contrast, metamers indistinguishable to human observers can be achieved when samples are statistically matched within all V2-sized pooling regions covering a large stimulus, mitigating the effect of the stimulus aperture”. It took a while for me to parse this sentence – if I understand correctly, the argument is that these metamers arise when statistics are matched within each aperture, rather than across an image which is then cropped to fill an aperture. If so, something like “when samples are statistically matched within each of the V2-sized pooling regions covering a large stimulus” might be more clear.

Finally, the revised methods state on line 397 that “Stimuli presented at 8 and 16 degrees eccentricity were shown at 40 and 20 pixels/degree”. I would suspect differences in viewing distance to achieve this, though line 394 states this was a constant 46cm. Was this achieved then through changes in monitor resolution? A touch more detail is required for this to be clear.

Reviewer #1

The authors addressed all our concerns to satisfaction. We find the new manuscript improved, in particular the new simulations and discussion of the requirement of normalization and surround suppression to observe the effects.

Thank you for your comments, and your suggestions for improving discussion of the issues you mention.

Reviewer #2

this is a very nice paper

Thank you.

Reviewer #3

The revised manuscript by Ziemba & Simoncelli presents a set of improvements to what was already a fascinating manuscript. The clarifications on model details have greatly improved the clarity of its operation, the new supplementary figures add details to help better understand both the data and the model, and revised text clarifies the position of these findings in the broader literature. I have no further major issues with the manuscript as it stands, though there are a few very minor details the authors may wish to clarify.

Thank you very much for your comments. We agree your suggestions helped us to significantly improve the manuscript and its clarity.

In the response to reviews the authors say that “As such, we don’t see that the phenomenon of crowding could provide an explicit prediction of our results”. However, line 169 of the revised manuscript states that “...populations of neurons in areas V2 and V4 are likely candidates for the locus of compulsory pooling giving rise to the opposing effects of stimulus size”. I take it that compulsory pooling is an apt description of the model processes here, but given the wide usage of this term to refer to models of crowding (following Parkes et al, 2001), a slightly different term would be ideal here, unless that equivalence is the authors’ intention.

This is a good point, and we have changed the text to read: “...populations of neurons in areas V2 and V4 are likely candidates for the locus of neural selectivity and invariance giving rise to the opposing effects of stimulus size”.

Regarding the relation to Wallis et al (2019), the authors' response states that "This is a somewhat intricate argument, and tangential to the current paper, so we allude to it only indirectly in the Discussion paragraph about the choice of statistical features". This is generally fine, though the phrasing in the discussion is a little too indirect to be clear. Line 243 states "In contrast, metamers indistinguishable to human observers can be achieved when samples are statistically matched within all V2-sized pooling regions covering a large stimulus, mitigating the effect of the stimulus aperture". It took a while for me to parse this sentence – if I understand correctly, the argument is that these metamers arise when statistics are matched within each aperture, rather than across an image which is then cropped to fill an aperture. If so, something like "when samples are statistically matched within each of the V2-sized pooling regions covering a large stimulus" might be more clear.

We have modified the sentence as suggested and agree it makes the point more clearly.

Finally, the revised methods state on line 397 that "Stimuli presented at 8 and 16 degrees eccentricity were shown at 40 and 20 pixels/degree". I would suspect differences in viewing distance to achieve this, though line 394 states this was a constant 46cm. Was this achieved then through changes in monitor resolution? A touch more detail is required for this to be clear.

We have added a note here specifying that we presented the images at a larger size by upsampling by factors of 2 and 4. Viewing distance and monitor resolution were the same for all experiments.